# TRUSTWORTHY DATASET PROOF: CERTIFYING THE AUTHENTIC USE OF DATASET IN TRAINING MODELS FOR ENHANCED TRUST

## ABSTRACT

In the realm of deep learning, the veracity and integrity of the training data are pivotal for constructing reliable and transparent models. This study introduces the concept of Trustworthy Dataset Proof (TDP), which tackles the significant challenge of verifying the authenticity of training data as declared by trainers. Existing dataset provenance methods, which primarily aim at ownership verification rather than trust enhancement, often face challenges with usability and integrity. For instance, excessive operational demands and the inability to effectively verify dataset authenticity hinder their practical application. To address these shortcomings, we propose a novel technique termed Data Probe, which diverges from traditional watermarking by utilizing subtle variations in model output distributions to confirm the presence of a specific and small subset of training data. This model-agnostic approach improves usability by minimizing the intervention during the training process and ensures dataset integrity via a mechanism that only permits probe detection when the entire claimed dataset is utilized in training. Our study conducts extensive evaluations to demonstrate the effectiveness of the proposed data-drobe-based TDP framework, marking a significant step toward achieving transparency and trustworthiness in the use of training data in deep learning.

## 1    INTRODUCTION

Against the backdrop of the rapid development of deep learning technologies, the reliability and transparency of models are increasingly being scrutinized, and the authentic use of training data is the cornerstone of building effective and trustworthy models (Shayne, 2024; Anomalo, 2024; Aldoseri et al., 2023). However, the actual usage of training data is often based on the trainer's self-report, which is extremely difficult to verify in practice.

The authenticity and integrity of training data are significant security concerns in the field of deep learning, as even minor tampering with the training data can lead to significant changes in model behavior. For instance, numerous studies (Chen & Babar, 2024; Khaddaj et al., 2023; Mengara et al., 2024; Saha et al., 2020) have shown that attackers can embed a small number of backdoor samples in training data, which could be activated during the model's deployment, posing security risks. On the other hand, trainers may introduce illegal or non-compliant data to enhance the performance of the model without informing the users, especially in competitions or commercial applications, raising concerns regarding the transparency and fairness. For example, high-profile lawsuits have been initiated against major tech companies like Google and StabilityAI, where plaintiffs argue that their copyrighted or personal data were used without permission to train AI systems (Sabine, 2024).

In this paper, we define the problem of **T**rustworthy **D**ataset **P**roof **(TDP)** to formally describe this challenging task of verifying the integrity of training datasets, and further explore possible solutions. As illustrated in Fig. 1, unlike the widely studied concept of data provenance, which focuses on the ownership verification of a personal dataset, the TDP problem seeks to verify the authentic use of a trustworthy dataset by the trainer, thereby enhancing trust.

**Challenges:** Although existing dataset provenance techniques hold promise for addressing the TDP problem, they still face several formidable challenges. ❶ **Usability**: these techniques often fail to meet practical requirements, such as demanding that the trainer provide all details of the training

Figure 1: **The Overview of the Trustworthy Dataset Proof (TDP) Problem.** Compared to the widely studied data provenance techniques, the TDP presents significant differences in terms of dataset utilization and verification scope, making it a more challenging and novel problem.

process (Choi et al., 2024; Jia et al., 2021), requiring the training of a dedicated classifier for each verification request (Maini et al., 2021; Dziedzic et al., 2022), or utilizing a backdoor for watermarking (Adi et al., 2018; Tang et al., 2023), which could potentially be exploited maliciously (Mengara et al., 2024; Khaddaj et al., 2023). ❷ **Integrity**: existing technologies are inadequate to certify the integral use of the training dataset claimed by the trainers. In contrast, they can only verify whether the actual dataset used approximates the distribution of the claimed dataset (Maini et al., 2021; Dziedzic et al., 2022; Choi et al., 2024; Jia et al., 2021), or whether it contains specific samples from the claimed dataset (Adi et al., 2018; Tang et al., 2023).

**Motivations:** To address the challenges of utility, we propose a novel concept named ***Data Probe***. This technique, distinct from traditional watermarking, does not require a model to produce a pre-determined output. Instead, it leverages subtle distinctions in model outputs distribution to validate its existence. It is designed to be model-agnostic, thereby reducing constraints on the training process. For further tackling the integrity challenges, we bind the integrity of the training dataset to the ***data probe selection strategy***. This ensures that successful probe implantation and detection are contingent upon the use of the complete dataset for training, serving as a possible solution to TDP.

**Contributions:** The contributions of our study are concluded as follows:

- To the best of our knowledge, this is ***the first exploration*** of the challenging security problem associated with verifying the comprehensive and authentic use of the training dataset.
- We formalize this issue as a Trustworthy Dataset Proof (TDP) problem, analyze the challenges posed by existing technologies, and derive technical insights for potential solutions.
- We innovatively design a watermarking-like technique called Data Probe, which underpins a TDP framework that is highly available and capable of verifying the integrity of training data.
- Extensive evaluations demonstrate the effectiveness of our proposed methods, validating our approach in various experimental settings.

## 2 RELATED WORK

Existing research on the protection and traceability of training datasets primarily focuses on verifying the ownership of datasets Although this objective differs from the goal discussed in this paper, which is to verify the integrity of training datasets to enhance credibility, we review these studies to better understand the existing challenges and potential technical motivations, with a more detailed discussion to follow in the Sec. 4.

**Dataset watermarking.** It achieves ownership authentication by embedding backdoors into training datasets (Adi et al., 2018). Specifically, the owner of a dataset can select and modify a small subset of samples to serve as backdoor triggers, which can induce atypical outputs from the model. Once the model is trained on this dataset, the backdoor is automatically implanted. Subsequently, the copyright holder of the dataset can verify whether a suspected model exhibits the expected anomalous behavior by activating the trigger, thereby confirming the occurrence of the training. Building on this, many studies focus on enhancing the stealth and harmlessness of these backdoors. For instance, Tang et al. (2023) propose to construct a clean-label backdoor by applying adversarial perturbations to samples, which embeds a specific backdoor pattern without altering the labels of the samples.

**Membership and Dataset Inference.** Membership Inference (**MI**) (Shokri et al., 2017; Salem et al., 2019; Yeom et al., 2018; Song & Mittal, 2021) is an attack targeting the privacy of training datasets, but its technical approach can also be reversed for tracing training sets. MI exploits the phenomenon

of overfitting in deep learning (Yeom et al., 2018), whereby a model better memorizes samples from its training set, distinguishing them from non-training samples. For instance, Shokri *et al.*(Shokri et al., 2017) proposed training numerous shadow classifiers to reveal such distinctions. Other studies analyze various score metrics produced by the model when different samples are inputted, such as confidence (Salem et al., 2019), loss (Yeom et al., 2018), and entropy (Song & Mittal, 2021). To further enhance the stability of verification, Maini et al. (2021) proposed the Dataset-Inference (**DI**) technique, which actually conducts MI at the distribution level and utilizes the characteristic differences in the model's responses to training and test samples.

**Proof of Training Data.** Proof-of-Training-Data (**PoTD**) (Choi et al., 2024) is a protocol enabling a model trainer to assure a verifier of the specific training data responsible for generating a set of model weights, which can confirm both the quantity and type of data. The protocol mandates that the model trainer meticulously document and provide comprehensive details throughout the training process, referred to as a training transcript, including the dataset, training codes, hyperparameters, and intermediate checkpoints. The goal of PoTD is similar to our work. However, it focuses on ensuring a precise correspondence between the training processes and model parameters, but fails to verify the presence of subtle manipulations in the training dataset, as stated in their work limitations.

## 3 FORMALIZE TDP

### 3.1 PROBLEM DEFINITION

In the **T**rustworthy **D**ata **P**roof (**TDP**) problem, two key roles are identified: **model trainer** and **verifier**. The trainer is capable of utilizing a publicly credible dataset $\mathcal{D}$ to train a model $\mathcal{M}$. On the other hand, the verifier is tasked with providing a mechanism to verify whether $\mathcal{M}$ was trained using $\mathcal{D}$ in a reliable and trustworthy manner. We formally define two functions to implement TDP:

**Definition 1** (Trusted Training) `T-Train(`$\mathcal{D}$`) →` $\mathcal{M}, \mathcal{C}$

*Trusted Training is a training mechanism specified by verifier, denoted as* `T-Train()`. *It could be further represented as* `T-Train` $= \mathbb{T} \cup \mathbb{O}$, ***where*** $\mathbb{T}$ ***refers to general model training procedures, and*** $\mathbb{O}$ ***represents additional operations required by the verifier.*** *The trainer need to use* `T-Train()` *to train models to meet the requirements for subsequent trustworthy verification. Specifically, it performs* `T-Train(`$\mathcal{D}$`)` *to obtain a trained model* $\mathcal{M}$, *and at the same time, get a certificate* $\mathcal{C}$ *for the subsequent verification .*

**Definition 2** (Verification) `Verify(`$\mathcal{D}, \mathcal{M}, \mathcal{C}$`) →` $\{0, 1\}$

*Verification is the process by which the verifier provide the proof, denoted as* `Verify()`. *The trainer want to assert that* $\mathcal{M}$ ***is trained on*** $\mathcal{D}$. *He then provides the verifier with* $\mathcal{M}$ *and* $\mathcal{C}$ *obtained through* `T-Train`, *as well as the* $\mathcal{D}$ *to be validated.* `Verify()` *outputs 1 to indicate that it judges the model is indeed trained on D, while 0 indicates it is not.*

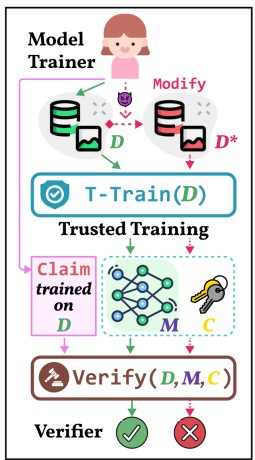

Figure 2: **The conceptual framework of TDP.**

We justify the rationality of the above definitions: ❶ **Generality:** the additional operations $\mathbb{O}$ and $\mathcal{C}$ can both be empty $\varnothing$. At this point, `T-Train` and `Verify` describe the most basic process of solving the TDP problem without any assumptions, making them universal. ❷ **Necessity:** according to the consensus in the relevant research field (Shayne, 2024; Aldoseri et al., 2023; Choi et al., 2024), due to the high-dimensional complexity of DL models and the non-convex optimization training process, it is difficult to directly verify the fact that $\mathcal{M}$ is trained on the complete $\mathcal{D}$ through $\mathcal{M}$ alone. Therefore, the verifier should be allowed to intervene and restrict the training process to enhance their ability during the verification process.

### 3.2 THREAT MODEL

In TDP, the verifier is generally served by trusted authorities to ensure the fairness and effectiveness of the verification process, and to defend against potential acts that may subvert the verification. Therefore, **the verifier is defined as the defender**. On the other hand, the trainer may maliciously exploit the TDP mechanism for unjust benefits, thus **the dishonest trainer is defined as the attacker**. In this section, we will analyze the goals and capabilities of each party separately.

**Defender's Goals.** The fundamental goal of the defender is to ensure the correctness of the verification (**G1**), accompanied by several performance-related goals (**G2, G3, G4**):

**G1. Fidelity.** If the trainer indeed carries out T-Train($\mathcal{D}$) → $\mathcal{M}, \mathcal{C}$, then the defender, executing Verify($\mathcal{D}, \mathcal{M}, \mathcal{C}$), must output 1 with high probability. In all other cases, output 0.

**G2. Low-Invasiveness.** In T-Train = $\mathbb{T} \cup \mathbb{O}$, it is advisable to minimize $\mathbb{O}$'s restrictions on $\mathbb{T}$, as excessive limitations can affect the trainer's flexibility in defining their own models and training processes, hindering the generalizability of the TDP mechanism.

**G3. Harmlessness.** Implementing TDP should not compromise the model's performance significantly. Specifically, the performance of the trained model $\mathcal{M}$ = T-Train($\mathcal{D}$) should be close to the performance of $\mathcal{M}_{\mathbb{T}} = \mathbb{T}(D)$, which is obtained using only the standard training process.

**G4. Efficiency.** The verification process should be computationally efficient.

**Defender's Capabilities.** Based on practical scenarios, we make the following assumptions and restrictions on the defender's capabilities. In T-Train = $\mathbb{T} \cup \mathbb{O}$, we assume that the additional operations required by the verifier, the $\mathbb{O}$, is *model-agnostic*. $\mathbb{O}$ will not modify the model architecture, nor obtain specific procedures or hyperparameters of $\mathbb{T}$. In Verify, we assume that the defender only has *black-box access* to the model $\mathcal{M}$ to be verified, and can only query the model's prediction probabilities, i.e., $y = \mathcal{M}(x)$, where $x$ is the input sample, $y \in \mathbb{R}^d$, and $d$ is the number of classes.

**Attacker's Goals.** The attacker is a dishonest model trainer, whose goal is to attempt to subvert the defender's fidelity goal **(G1)**, formally defined as follows:

> When the attacker use the modified dataset $\mathcal{D}^* \neq \mathcal{D}$ in trusted training, they would obtain T-Train($\mathcal{D}^*$) → $\mathcal{M}^*, \mathcal{C}^*$. But they purposely claim to the defender (verifier) that ***the model*** $\mathcal{M}^*$ ***is trained on*** $\mathcal{D}$, attempting to make Verify($\mathcal{D}, \mathcal{M}^*, \mathcal{C}^*$) → 1.

It is important to emphasize that the attacker's primary objective is to successfully complete the TDP. By obtaining the proof that *the model's training set is a credible dataset* $\mathcal{D}$, the attacker can gain the trust of users. As a result, the model $\mathcal{M}^*$ it publicly releases is more likely to be downloaded, deployed, and used. On this basis, the attacker attempts to 'hide' the fact that it actually tampered with the training dataset for training.

**Attacker's Capabilities.** We assume that the attacker can only modify the dataset $\mathcal{D}$ but cannot manipulate the verifier's additional operations $\mathbb{O}$ in T-Train. The reasonableness of this assumption is consistent with the discussion in problem definition (Sec. 3.1). The defender (verifier) needs to intervene in the model training with certain operations $\mathbb{O}$ to facilitate verification. Once the attacker can manipulate $\mathbb{O}$, they can easily neutralize various verification mechanisms of the defender. Therefore, we reasonably limit the capabilities of the attacker, requiring them to at least fully execute T-Train. In other words, any modifications made by the attacker to $\mathcal{M}$ must go through T-Train.

## 4 CHALLENGES AND MOTIVATIONS

### 4.1 CHALLENGES FOR CURRENT TECHNIQUES

Based on the formal definition of TDP in Sec. 3, we utilize representative existing dataset provenance technologies, including Watermarking, MI/DI, and PoTD, to propose some hypothetical solutions for TDP and analyze the challenges involved. Detailed definitions and analyses of each approach are elaborated in the Appendix A, from which we conclude that the application of existing schemes to TDP primarily confronts challenges in two aspects: *integrity* and *usability*.

**Challenge 1: Intergrity**. All existing schemes fail to ensure the integrity of the training dataset $\mathcal{D}$. Specifically, if an attacker claims to have used $\mathcal{D}$ for training but actually employs $\mathcal{D}^*$, current schemes can only verify whether $\mathcal{D}^*$ approximates the distribution of $\mathcal{D}$ (MI/DI, PoTD), or whether $\mathcal{D}^*$ contains several specific samples from $\mathcal{D}$ (Watermarking). However, none of these approaches can confirm that $\mathcal{D}^* \neq \mathcal{D}$.

This disadvantage fundamentally stems from differences in the definition of the threat model, which is also detailed in Fig. 1. In most of the aforementioned works, the party holding the data primarily aims at *ownership verification*. Therefore, they are typically regarded as trustworthy or defenders, with no incentive to actively alter their dataset. However, in TDP, the main goal of the data holder

and user is to *enhance trust*, which might lead them to actively modify the dataset while still claiming to use an authentic dataset to gain undue benefits. This disparity is the root of the vulnerabilities in existing methods, rendering them unable to meet the fundamental goal of fidelity (**G1**) in TDP.

**Challenge 2: Usability**. Some existing techniques may exhibit deficiencies in usability within practical scenarios. For instance, the PoTD scheme does not meet the defender's goal: low-invasiveness (**G2**), as it requires accessing all information from the model trainer during training. DI, to some extent, fails to satisfy the efficiency goal (**G4**), since it may necessitate training a dedicated classifier for each verification request. The watermarking approach could potentially compromise the defender's goal of harmlessness (**G3**), as watermarks that utilize backdoors might be maliciously exploited, thereby introducing inherent security risks.

## 4.2 MOTIVATIONS AND INSIGHTS

**Tackling usability challenge**. Among various approaches, we consider the concept of watermarking to exhibit the highest usability. It imposes minimal constraints during the training phase and offers the highest efficiency during verification, with only requiring black-box access to the model under test. However, its shortcomings originate from the use of backdoors (Adi et al., 2018), typically requiring the model to produce anomalous outputs when presented with specific trigger samples, which often leads to degradation in model performance or security risks (Mengara et al., 2024)

We aim to relax assumptions regarding watermark capabilities to enhance usability while ensuring sufficient capability to determine the occurrence of watermark. We also draw inspiration from MI, noting that the membership of samples could be determined based on discrepancies in model output. Consequently, we introduce a new concept: **Data Probe**, for conducting watermarking operations. Its formal definition is as follows:

**Definition 3** (Data Probe). *Data probe is defined as a small subset of samples, $\mathbf{x}_p$, within the training dataset $\mathcal{D}$. Based on this, the training dataset can be divided into two parts: $\mathcal{D} = \{\mathbf{x}_p \cup \mathbf{x}_{np}\}$, where $\mathbf{x}_{np}$ refers to 'non-probe' samples. After training the model $\mathcal{M}$ on $\mathcal{D}$, we expect a noticeable difference in the output distribution of the model when inputting data from these two subsets, i.e.,*

$$\Pr(\mathbf{y}|\mathbf{x}_p; \mathcal{M}) \neq \Pr(\mathbf{y}|\mathbf{x}_{np}; \mathcal{M})$$

Data probe can be considered as a weakened version of a backdoor, it only necessitates a discernible difference in the output compared to that from non-probe inputs $\mathbf{x}_{np}$, such as a slight increase in prediction confidence, but not a 'directed' output. From the perspective of MI, although $\mathbf{x}_p$ and $\mathbf{x}_{np}$ both belong to the training set, we expect $\mathbf{x}_p$ to behave more like a 'special member' of the dataset.

**Tackling intergrity challenge.** Simple data probe selection and implanting strategies may still fall into the dilemmas encountered by watermarking schemes. Hence, our insight is to ***bind the integrity of the training dataset with the data probe selection strategy***, ensuring that successful probe implantation and detection can only occur when the dataset is used in its entirety for training.

## 5 IMPLEMENT TDP

**Conceptual Overview.** Our proposed data-probe-based TDP framework is illustrated in Fig. 3 and formally described in Algorithm 1. Relevant functions are denoted with the subscript DP.

In T-Train$_{\mathsf{DP}}$, a pseudo-random mechanism ProbeSelect is introduced to select the data probe $\mathbf{x}_p$. It needs to perform a keyed-hash on training dataset $\mathcal{D}$ based on a user-specific key $\mathbf{k}$. Next, operation $\mathbb{O}_{\mathsf{DP}}$ is exerted on the selected probe to facilitate its 'implantation' into the model $\mathcal{M}$ during training. Subsequently, the trainer can carry out the normal training process $\mathbb{T}$ and submit the key $\mathbf{k}$ as a certificate $\mathcal{C}$ for verification.

In Verify$_{\mathsf{DP}}$, the verifier first reproduces the pseudo-random process ProbeSelect based on $\mathcal{C}$ to select data probe $\mathbf{x}_p$ and non-probe $\mathbf{x}_{np}$ from $\mathcal{D}$ claimed by the trainer. He then calculates the scores $\mathbf{s}_p, \mathbf{s}_{np}$ through a function ProbeScore to measure the difference in the output distribution of these two groups of data in model $\mathcal{M}$. If there is a noticeable difference between $\mathbf{s}_p$ and $\mathbf{s}_{np}$, it is considered that the data probe $\mathbf{x}_p$ has been detected, authenticating the genuine use of the training dataset $\mathcal{D}$.

Figure 3: **The Conceptual Overview of the Proposed TDP Framework based on Data Probe.**

$\text{Verify}_{\text{DP}}$ satisfies the main objective of the defender: fidelity **(G1)**. The principle lies in that, if the trainer indeed use $\mathcal{D}$ for training $\mathcal{M}$, the data probe $\mathbf{x}_p$ selected and implanted in $\text{T-Train}_{\text{DP}}$ will be completely consistent with the probe calculated in $\text{Verify}_{\text{DP}}$. Assuming that the data probe can correctly perform the function defined in Definition 3, the verifier can reliably certify its claim. Conversely, if the trainer makes any minor modifications to $\mathcal{D}$, as shown in the lower part of Fig. 3, the implanted data probe $\mathbf{x}_p^*$ will not correspond with $\mathbf{x}_p$, thus failing the validation.

**Probe Selection.** The probe selection rule is based on pseudo-randomness, which utilizes the *uniqueness* of hash functions (Rivest, 1992). Initially, it calculates a hash of the complete dataset $\mathcal{D}$ to obtain a unique hash value. Subsequently, we use this hash value as a random seed to randomly select a small subset of samples from the training dataset as data probe $\mathbf{x}_p$, as described in Lines 3-4 of Algorithm 1. Clearly, $\mathcal{D}$ and $\mathbf{x}_p$ are bound together, and any minor modification of $\mathcal{D}$ will result in changes to the hash value, which in turn leads to changes in the selection of data probe $\mathbf{x}_p$.

However, simple hash calculations pose security risks. The same dataset will yield the same data probe for all users, which could be easily exploited by attackers. Therefore, we introduce the user-specific key $\mathbf{k}$ and replace the hash with keyed-hash, such that different users using the same trusted dataset (such as CIFAR10 (Krizhevsky & Hinton, 2009)) will generate different data probes.

**Probe Implantation.** Reflecting on the functionality of data probe described in Definition 3, our goal is to elicit a special response from the trained model $\mathcal{M}$ to the data probe $\mathbf{x}_p$. At the same time, we also need to fully consider the usability of the scheme. $\mathbb{O}_{\text{DP}}$ does not need to change the architecture of model $\mathcal{M}$, nor does it need to obtain the hyperparameters in training process $\mathbb{T}$. Therefore, we have determined that $\mathbb{O}_{\text{DP}}$ should only perform data-level operations on data probe $\mathbf{x}_p$, without needing to change the normal training process $\mathbb{T}$.

Table 1: **Data Probe Types and Principles.** PS represents the generic scoring process, calculating the probe score $\mathbf{s}_p$ and non-probe score $\mathbf{s}_{np}$. $\mathbf{l}_p$ and $\mathbf{l}_{np}$ denote the ground-truth labels. $\mathbf{l}_p^t$ and $\mathbf{l}_{np}^t$ are targeted labels defined in TP.

| Type | $\mathbf{s}_p$ | $\mathbf{s}_{np}$ | Expectation |
|------|------|------|------|
| PP | $\text{PS}(\mathcal{M}, \mathbf{x}_p, \mathbf{l}_p)$ | $\text{PS}(\mathcal{M}, \mathbf{x}_{np}, \mathbf{l}_{np})$ | $\mathbf{s}_p^{\text{PP}} > \mathbf{s}_{np}^{\text{PP}}$ |
| AP | $\text{PS}(\mathcal{M}, \mathbf{x}_p, \mathbf{l}_p)$ | $\text{PS}(\mathcal{M}, \mathbf{x}_{np}, \mathbf{l}_{np})$ | $\mathbf{s}_p^{\text{AP}} < \mathbf{s}_{np}^{\text{AP}}$ |
| UP | $\text{PS}(\mathcal{M}, \mathbf{x}_p, \mathbf{l}_p)$ | $\text{PS}(\mathcal{M}, \mathbf{x}_{np}, \mathbf{l}_{np})$ | $\mathbf{s}_p^{\text{UP}} < \mathbf{s}_{np}^{\text{UP}}$ |
| TP | $\text{PS}(\mathcal{M}, \mathbf{x}_p, \mathbf{l}_p^t)$ | $\text{PS}(\mathcal{M}, \mathbf{x}_{np}, \mathbf{l}_{np}^t)$ | $\mathbf{s}_p^{\text{TP}} > \mathbf{s}_{np}^{\text{TP}}$ |

Inspired by existing dataset provenance research, we have developed four different types of data probe: Prominent Probe **(PP)**, Absence Probe **(AP)**, Untargeted Probe **(UP)**, and Targeted Probe **(TP)**. We summarize the principles of various probes in Tab. 1 and detail their concepts and implementations in this section. The unique operations of different types of probes are marked by superscripts. Besides, the performance and characteristics of the four probes are compared in Sec. 6.2.

❶ **Prominent Probe (PP).** 'Prominent' means the model's response to the data probe $\mathbf{x}_p$ is more significant compared to non-probe $\mathbf{x}_{np}$. In principle, we aim to make the model more overfit on the $\mathbf{x}_p$, providing *more confident* scores. $\mathbb{O}_{\text{DP}}^{\text{PP}}$ can be implemented through the built-in data sampling mechanism of the deep learning computation library, by assigning a higher weight to the $\mathbf{x}_p$. The $\mathbf{x}_p$ is therefore more likely to be selected during the training, resulting in better fitting to the $\mathbf{x}_p$.

❷ **Absence Probe (AP).** The principle of AP is exactly opposite to that of PP. To maximize the insignificance, we consider an extreme case where the probe weight is 0, meaning the model has never encountered $\mathbf{x}_p$ during training. Then the trained model $\mathcal{M}$ should provide *less confident* scores for $\mathbf{x}_p$. $\mathbb{O}_{\text{DP}}^{\text{AP}}$ can be implemented in a similar manner to $\mathbb{O}_{\text{DP}}^{\text{PP}}$ by setting the probe weight to 0.

❸ **Untargeted Probe (UP).** Untargeted Probe utilizes the concept of data poisoning (Adi et al., 2018; Mengara et al., 2024), where $\mathbb{O}_{\mathsf{DP}}^{\mathsf{UP}}$ assigns random, incorrect labels to the probe $\mathbf{x}_p$. In this case, we expect the model to be *less confident* in its predictions for the probe $\mathbf{x}_p$. For instance, a decrease is witnessed in the predicted probabilities for their ground truth classes.

❹ **Targeted Probe (TP).** In contrast to UP, Targeted Probe is where $\mathbb{O}_{\mathsf{DP}}^{\mathsf{TP}}$ uniformly assigns a random label, marked as $l^t$, to the probe. In this case, we expect the trained model to be *more confident* in predicting the $\mathbf{x}_p$ as class $l^t$. Assuming the number of probe and non-probe are $N_p$ and $N_{np}$, we construct two sets of labels: $\mathbf{l}_p^t = \{l^t\}^{N_p}$ and $\mathbf{l}_{np}^t = \{l^t\}^{N_{np}}$ and obtain scores as stated in Tab. 1.

**Probe Score Calculation.** The detection of data probe relies on comparing the saliency scores of probe $\mathbf{s}_p$ and non-probe $\mathbf{s}_{np}$. In this section, we will demonstrate possible scoring methods. Inspired by the techniques for assessing sample saliency adopted in the MI work, we propose the following four possible methods: confidence-based, loss-based, entropy-based, and modified-entropy-based.

According to the aforementioned `ProbeScore` function protocol, the inputs include a model $\mathcal{M}$, a set of data $\mathbf{x}$, and a set of labels $\mathbf{l}$. For the sake of brevity in expression, we describe the score calculation method for individual samples $x \sim \mathbf{x}$, without distinguishing between probe and non-probe, as they actually use the same calculation method. Besides, we convert the corresponding label $l \sim \mathbf{l}$ into the index of the model's output for convenience in expression, i.e., when $l$ indicates that $x$ belongs to class $y$, we use $\mathcal{M}(x)_y$ to represent the $\mathcal{M}$'s prediction probability for $x$ being in that class.

❶ **Confidence-based score (`Conf`)** For a more *significant* sample during training, the model should make predictions with higher confidence in it (Salem et al., 2019). we define the confidence-based score as: $\boxed{\mathsf{Conf}(\mathcal{M}, x) = \max(\mathcal{M}(x))}$ which will be directly return as the probe score.

❷ **Loss-based score (`Loss`)** For a more *significant* sample during training, the model should has a lower prediction loss on it (Yeom et al., 2018). We mark the loss function, such as the cross-entropy, as $\mathcal{L}$, and we define the loss-based score as: $\boxed{\mathsf{Loss}(\mathcal{M}, x, l) = \mathcal{L}(\mathcal{M}(x), l)}$. The more significant the sample $x$, the smaller the `Loss`. In order to make the return values of `ProbeScore` continuous, where larger values represent more significance, we return **-`Loss`** as the probe score.

❸ **Entropy-based score (`Entr`)** For a more *significant* samples during training, the model's prediction on it should be close to the one-hot encoded label, i.e., its entropy will be close to $0$ (Salem et al., 2019). We define the entropy-based score as: $\boxed{\mathsf{Entr}(\mathcal{M}, x) = -\sum_i \mathcal{M}(x)_i \log(\mathcal{M}(x)_i)}$. Similar to `Loss`, we return **-`Entr`** as the probe score.

❹ **Modified-entropy-based score (`Mentr`)** It is an enhanced version of `Entr` by considering the ground-truth label $l$ (Song & Mittal, 2021). We define the modified-entropy-based score as: $\boxed{\mathsf{Mentr}(\mathcal{M}, x, l) = -(1 - \mathcal{M}(x)_y)\log(\mathcal{M}(x)_y) - \sum_{i \neq y} \mathcal{M}(x)_i \log(\mathcal{M}(x)_i)}$. Similar to `Loss`, we return **-`Mentr`** as the probe score.

**Probe Detection Metric.** According to the definition of Data Probe, probe scores are expected to differ in distribution from non-probe scores. Therefore, we aggregate the aforementioned sample-level scores into distribution-level metrics for probe detection. The following two metrics are employed: ❶ **P**robe **S**aliency **AUC** (**PSA**). It is a new metric proposed in this work. It utilizes probe scores to plot the ROC curve, further calculating the Area Under ROC curve (AUC) as the metric. The greater the PSA exceeds 0.5, the more it indicates that probe scores are separable from non-probe scores. ❷ Statistical test **p-value** (**pV**). It has been widely adopted in previous works for measuring distribution differences. A pV less than a certain level of significance, such as 0.1, indicates a substantial difference in score distributions, signifying that the probe has been detected. We detail the principles of these two metrics and their calculation methods for each probe in Appendix B.

## 6 EVALUATIONS

**Overview.** In the evaluations, we aim to investigate the following four research questions (RQs):

**RQ1.** Whether various types of data probes can effectively verify the integrity of the dataset ?

**RQ2.** How many probes need to be implanted during training to achieve the verification ?

**RQ3.** How do different probe score calculation strategies impact the effectiveness of detection ?

**RQ4.** How robust is the verification mechanism when attackers launch adaptive attacks ?

Table 2: **Comprehensive Probe Perfomance Evaluations.** Best performance of PSA and pV under each setting are hilighted as BLUE and RED. And the metric under probe-mismatch cases are marked as GRAY. Scores are reported as % except for pV.

| | Ori. | PP | | | | | AP | | | | | UP | | | | | TP | | | | |
|---|---|---|---|---|---|---|---|---|---|---|---|---|---|---|---|---|---|---|---|---|---|
| | acc | acc | PSA↑ | PSA* | pV↓ | pV* | acc | PSA↑ | PSA* | pV↓ | pV* | acc | PSA↑ | PSA* | pV↓ | pV* | acc | PSA↑ | PSA* | pV↓ | pV* |
| CIFAR-10 | | | | | | | | | | | | | | | | | | | | | |
| ResNet18 | 85.22 | 85.19 | 56.15 | 50.58 | $\mathbf{10^{-21}}$ | 0.35 | 85.08 | 52.14 | 49.97 | 0.01 | 0.42 | 84.68 | 57.09 | 49.60 | 0.02 | 0.55 | 84.92 | **59.22** | 50.18 | $10^{-12}$ | 0.47 |
| MobileNet | 84.55 | 84.12 | 55.95 | 49.60 | $\mathbf{10^{-16}}$ | 0.54 | 84.64 | 52.59 | 50.09 | 0.02 | 0.39 | 84.25 | 57.02 | 49.99 | $10^{-4}$ | 0.37 | 84.43 | **57.17** | 50.16 | $10^{-7}$ | 0.48 |
| ShuffleNet | 84.69 | 83.06 | 57.20 | 50.41 | $\mathbf{10^{-43}}$ | 0.50 | 82.57 | 52.66 | 49.67 | 0.03 | 0.41 | 86.77 | **59.96** | 49.53 | $10^{-5}$ | 0.39 | 82.83 | 57.87 | 49.78 | $10^{-9}$ | 0.59 |
| DenseNet | 86.32 | 85.27 | 55.58 | 50.11 | $\mathbf{10^{-27}}$ | 0.51 | 86.18 | 52.60 | 49.42 | 0.08 | 0.55 | 83.04 | 57.63 | 48.95 | $10^{-3}$ | 0.35 | 86.49 | **59.51** | 49.81 | $10^{-9}$ | 0.56 |
| SVHN | | | | | | | | | | | | | | | | | | | | | |
| ResNet18 | 92.03 | 91.86 | 51.28 | 49.25 | $\mathbf{10^{-11}}$ | 0.65 | 91.88 | 50.68 | 50.84 | 0.29 | 0.28 | 92.10 | 53.38 | 51.04 | 0.27 | 0.24 | 92.15 | **53.85** | 50.59 | $10^{-4}$ | 0.42 |
| MobileNet | 91.98 | 91.43 | 51.17 | 49.33 | $\mathbf{10^{-6}}$ | 0.64 | 91.67 | 50.74 | 50.41 | 0.28 | 0.56 | 91.81 | 52.79 | 50.48 | 0.26 | 0.29 | 91.91 | **52.83** | 50.40 | 0.01 | 0.41 |
| ShuffleNet | 92.21 | 91.50 | 51.32 | 49.44 | $\mathbf{10^{-4}}$ | 0.78 | 91.66 | 51.01 | 50.44 | 0.30 | 0.35 | 92.41 | 52.84 | 50.72 | 0.35 | 0.21 | 91.75 | **53.56** | 50.23 | $10^{-3}$ | 0.50 |
| DenseNet | 92.45 | 91.89 | 51.40 | 49.32 | $\mathbf{10^{-3}}$ | 0.67 | 92.37 | 51.21 | 50.72 | 0.26 | 0.29 | 92.00 | 53.29 | 50.58 | 0.19 | 0.25 | 92.03 | **53.44** | 50.46 | $10^{-3}$ | 0.42 |
| CIFAR-100 | | | | | | | | | | | | | | | | | | | | | |
| ResNet18 | 64.62 | 63.57 | 69.20 | 50.06 | $\mathbf{10^{-99}}$ | 0.66 | 63.83 | 58.07 | 49.29 | $10^{-10}$ | 0.48 | 63.95 | 71.28 | 49.65 | $10^{-27}$ | 0.62 | 64.39 | **79.83** | 50.67 | $10^{-92}$ | 0.40 |
| MobileNet | 62.45 | 61.17 | **69.36** | 50.72 | $\mathbf{10^{-99}}$ | 0.40 | 61.68 | 56.75 | 49.03 | $10^{-7}$ | 0.64 | 62.30 | 63.42 | 48.89 | $10^{-11}$ | 0.63 | 62.35 | 69.04 | 50.09 | $10^{-38}$ | 0.51 |
| ShuffleNet | 60.22 | 59.40 | 70.13 | 50.33 | $\mathbf{10^{-99}}$ | 0.51 | 59.86 | 57.39 | 50.02 | $10^{-8}$ | 0.53 | 60.30 | 63.27 | 49.49 | $10^{-12}$ | 0.69 | 60.48 | **71.14** | 50.66 | $10^{-48}$ | 0.31 |
| DenseNet | 64.42 | 63.44 | 69.21 | 50.40 | $\mathbf{10^{-99}}$ | 0.57 | 64.39 | 57.67 | 49.81 | $10^{-10}$ | 0.65 | 64.33 | 68.12 | 49.27 | $10^{-12}$ | 0.66 | 64.61 | **77.56** | 50.64 | $10^{-83}$ | 0.42 |
| Tiny-ImageNet-200 | | | | | | | | | | | | | | | | | | | | | |
| ResNet18 | 53.00 | 51.03 | **76.26** | 50.17 | $\mathbf{10^{-54}}$ | 0.47 | 51.83 | 62.89 | 50.30 | $10^{-29}$ | 0.41 | 53.35 | 49.82 | 50.61 | 0.44 | 0.38 | 53.18 | 49.94 | 49.21 | 0.63 | 0.68 |
| MobileNet | 51.06 | 49.41 | **75.38** | 49.47 | $\mathbf{10^{-75}}$ | 0.62 | 50.12 | 57.35 | 50.44 | $10^{-6}$ | 0.46 | 50.83 | 50.20 | 50.56 | 0.41 | 0.42 | 51.00 | 49.87 | 49.68 | 0.59 | 0.63 |
| ShuffleNet | 50.11 | 47.85 | **76.02** | 49.85 | $\mathbf{10^{-78}}$ | 0.44 | 48.47 | 57.99 | 50.31 | $10^{-9}$ | 0.51 | 50.04 | 49.58 | 50.20 | 0.45 | 0.55 | 49.81 | 49.38 | 50.15 | 0.61 | 0.50 |
| DenseNet | 55.26 | 53.16 | **74.63** | 50.02 | $\mathbf{10^{-55}}$ | 0.45 | 53.86 | 60.51 | 50.00 | $10^{-20}$ | 0.55 | 55.72 | 49.58 | 50.02 | 0.40 | 0.51 | 55.59 | 50.05 | 49.63 | 0.52 | 0.64 |

## 6.1 SETUP

**Datasets:** Four datasets are adopted: **CIFAR-10** (Krizhevsky & Hinton, 2009), **SVHN** (Netzer et al., 2011), **CIFAR-100** (Krizhevsky & Hinton, 2009), and **Tiny-ImageNet-200** (Le & Yang, 2015).

**Models:** We employed various architectures: **ResNet18** (He et al., 2016), **MobileNet** (Howard et al., 2017), **ShuffleNet** (Zhang et al., 2018), and **DenseNet** (Huang et al., 2017). These models were chosen to ensure a broad evaluation of performance across different architectural dynamics.

**Metrics:** We utilized two metrics: **PSA** and **p-value(pV)**, which have been introduced in Sec. 5, for probe detection and effectiveness evaluation. The significance threshold for pV is set to 0.1.

For further specifics, such as hyperparameters, please refer to the Appendix C.

## 6.2 EVALUATION RESULTS

**Comprehensive performance comparison (RQ1).** We comprehensively evaluated the performance of various architectures when trained on different datasets using the four types of data probes proposed for TDP. The results are displayed in the Tab. 2. All experiments were repeated five times, and the mean values were recorded. We tested the accuracy of models trained directly without any probes as a baseline for comparison, denoted as "**Ori.**". For each type of probe, in addition to training accuracy, we recorded two metrics when the declared dataset matched the actual training dataset: **PSA** and p-value (**pV**). Additionally, we tested metrics for mismatched declared and training models as a contrast (**PSA***  and **pV***). In each experimental group, we selected 1% of the training set to serve as probes. All probe scores were calculated using `Mentr`.

Through extensive experimental comparisons, we summarize the following results: ❶ **The implantation of data probes has almost no impact on the performance of the original model.** As shown by the accuracy metrics in Tab. 2, the performance of models with data probes deviates minimally from the original performance, with the majority of variations within an acceptable range (<±1%). This meets the defender's goal of harmlessness (**G3**). ❷ **Data probes generally achieve the task of verifying dataset completeness.** Specifically, for the PSA metric, we expect it to approach 0.5 when probes do not match, and significantly exceed 0.5 when they do match. For the pV, values below 0.1 indicate that probes have been detected with high confidence, while in cases of mismatch, we expect the p-value to be as high as possible. It can be seen that most probes meet the aforementioned requirements across various training sets and models, successfully implementing TDP. Additionally, we observed that among the four types of probes, PP exhibits high significance in the p-value metric, while TP demonstrates a clear advantage in the PSA metric.

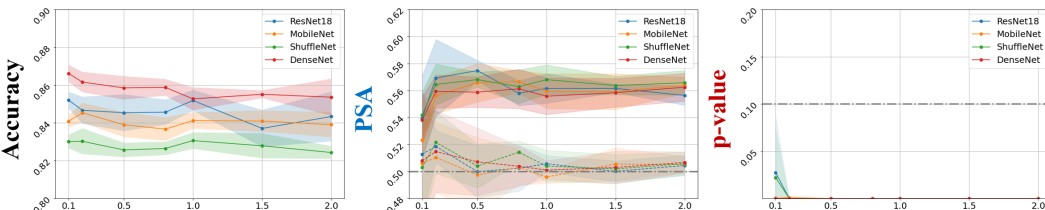

Figure 4: **Evaluations of the Impact of Probe Implant Quantity** We use a gray dashed line to mark two critical reference values: PSA = 0.5 and pV = 0.1. In PSA, lighter-colored dashed lines are non-probe scores. Here we present the results of PP, with the complete results shown in Fig. 7.

**Impact of probe quantity (RQ2).** We assessed how the performance metrics of TDP vary with an increase in the number of probes, ranging from 0.1% to 2% of the training set, as shown in Fig. 4. We trained various models on the CIFAR-10 dataset. All experiments were repeated five times, and the mean values as well as the standard deviation were recorded and displayed. The probe scores were also calculated using Mentr. From the evaluation results, we can conclude that **selecting approximately 1% of the training set samples is sufficient to reliably implement TDP.** As the number of implanted samples increases, the performance of the model may experience a negligible decline, while the metrics of PSA and p-value tend to stabilize. We believe that using about 1% of the training data as probes effectively achieves TDP, representing an appropriate trade-off between model verifiability and performance.

**Impact of probe score calculation methods (RQ3).** We analyzed the viability of various possible probe score calculation methods introduced in Sec. 5, as displayed in Fig. 5. We trained ShuffleNet on CIFAR-10 using various default settings for different probes and calculated the probe scores using four different methods. All experiments were repeated five times, and the mean values were reported. From the experimental results, **we found that Mentr exhibits the best generalizability and detection effectiveness**. It effectively detected data probes when applied across all four probe schemes. Additionally, we discovered that Conf and Entr are completely inapplicable to TP, as TP relies on inducing samples to point towards a specific label, whereas the calculations for Conf and Entr are independent of the label.

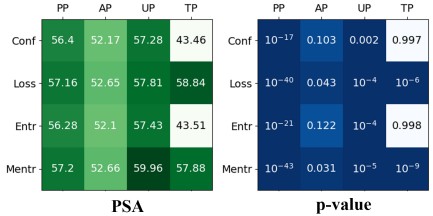

Figure 5: **Evaluations of Various Probe Score Calculation Methods.**

**Robustness to adaptive attackers (RQ4).** We evaluate a worst-case security scenario in which the attacker knows the key $\mathbf{k}$ used in the probe selection process and the specific type of probe employed. Specifically, the attacker can replicate the probe $\mathbf{x}_p$ corresponding to $\mathcal{D}$ after training the model $\mathcal{M}^*$ using $\mathcal{D}^*$, and attempt to illegitimately pass the verification for $\mathcal{D}$ by embedding the probe into the model. We therefore designed four probe forging attacks for each probe, with the underlying principles detailed in Appendix D, denoted with the prefix **F** for "Forged". Evaluations adopt ShuffleNet on CIFAR-10. We trained 10 different models randomly and designated 10 different target probes for each model, resulting in a total of 100 experimental groups per attack. We recorded the mean and standard deviation of various metrics before and after the attack. Additionally, we set the PSA threshold at 0.51 and the p-value at 0.1, and documented the Attack Success Rate (**ASR**).

The complete training results are displayed in Tab. 3. We found that **all probe schemes, except for PP, exhibit a certain degree of robustness.** This indicates that in most cases, it is challenging for attackers to easily deceive the verification mechanism through forged probes without compromising model performance. Additionally, we observed certain changes in metrics after the attacks, revealing potential security risks. Therefore, keeping the user's key $\mathbf{k}$ hidden from the users, such as by implementing it through a server API, might be a solution to enhance robustness.

Table 3: **Evaluations of the Robustness to Adaptive Probe Forging Attacks**. Metrics before and after attack are shown in GRAY and BOX.

| Attacks | Acc | PSA | | pV | |
| | | score | ASR | score | ASR |
| --- | --- | --- | --- | --- | --- |
| FPP | 84.0±0.4 | 50.1±1.7 | 62% | 0.28±0.25 | 67% |
| | 84.0±0.5 | 54.5±1.6 | | $10^{-6} \pm 10^{-5}$ | |
| FAP | 83.6±0.3 | 50.7±1.7 | 14% | 0.45±0.27 | 32% |
| | 82.3±0.5 | 51.4±1.7 | | 0.20±0.21 | |
| FUP | 83.6±0.3 | 50.7±1.7 | 5% | 0.49±0.27 | 0% |
| | 83.0±0.4 | 51.2±1.7 | | 0.49±0.28 | |
| FTP | 83.6±0.3 | 50.8±1.5 | 9% | 0.37±0.26 | 10% |
| | 82.7±0.5 | 51.4±1.5 | | 0.27±0.22 | |

| Description | Sample | Original Trigger Score | Watermark | | Dataset Inference | | Data Probe (Ours) | | | |
|---|---|---|---|---|---|---|---|---|---|---|
| | | | Verify Result | Trigger Score | Verify Result | Trigger Score | Verify Result | Trigger Score | HASH (train) | HASH (test) |
| Automobile ↓ Deer |  | 99.82 Deer | PASSED 1.000 | 84.24 Deer | PASSED $2.3 \times 10^{-7}$ | 99.82 Deer | BLOCKED 0.85 | 98.06 Deer | 90db8061ae4 0dcbd47f31bf 17e173651 | 9504b703430 662aad7aa7c 4a9f0333b2 |

Figure 6: **Case Studies for Comparison with Existing Techniques.** The scores for each validation are presented in GRAY, with watermarking representing the predicted probabilities, and DI and Data Probe indicating the p-values. HASH(train) shows the hash value computed on the tampered dataset during probe implantation, which differs from the HASH(test) obtained on the declared untampered dataset (CIFAR-10) during testing. More results are exhibited in Fig. 8 in the Appendix.

# 7 CASE STUDY

We conducted several case studies to validate the efficacy of the proposed Data-Probe-based TDP (DP) when genuine modifications occur within datasets. **PP** is adopted for evaluations. Meanwhile, we conducted comparative analyses with existing technologies, selecting two representative approaches: **Watermarking** (Tang et al., 2023) (WM) and **Dataset Inference** (Maini et al., 2021) (DI). Although they are not originally designed for the TDP mission, we adapted them using the hypothetical schemes proposed in Appendix A, utilizing their official open-source implementations. We employed CIFAR-10 and ResNet18 to test the verification results when an attacker claims to have trained on CIFAR-10, while subtly modifying it in training, which is expected to fail verification.

**Simulated modification.** We simulated two typical scenarios of dataset tampering: introducing additional data and embedding backdoors. For the former, we randomly selected a small proportion of samples, from 0.01% to 1%, and duplicated them. For the latter, we chose a small subset of samples and applied minimal noise (bounded by $l_\infty = 8/255$), adhering to the common configurations used in backdoor attacks. For

Table 4: **Case Study of Simulated Modifications.** Success rate (%) is shown in $\boxed{\text{BOX}}$ with scores in GRAY, consistent with Fig. 6.

| Method | Origin | Extra Data | | | Backdoor | | | Usability Eva. | |
|---|---|---|---|---|---|---|---|---|---|
| | | 0.01% | 0.10% | 1% | 0.01% | 0.10% | 1% | Sec. Risk | Time |
| WM | $\boxed{100}$ 1.0 | $\boxed{0}$ 1.0 | $\boxed{0}$ 1.0 | $\boxed{0}$ 0.99 | $\boxed{0}$ 0.99 | $\boxed{0}$ 0.99 | $\boxed{5}$ 0.97 | 35.60% | 8.8s |
| DI | $\boxed{100}$ $10^{-7}$ | $\boxed{0}$ $10^{-7}$ | $\boxed{0}$ $10^{-7}$ | $\boxed{0}$ $10^{-7}$ | $\boxed{0}$ $10^{-7}$ | $\boxed{0}$ $10^{-7}$ | $\boxed{0}$ $10^{-7}$ | - | 58min |
| DP(**Ours**) | $\boxed{100}$ $10^{-6}$ | $\boxed{100}$ 0.53 | $\boxed{100}$ 0.58 | $\boxed{100}$ 0.46 | $\boxed{95}$ 0.54 | $\boxed{100}$ 0.50 | $\boxed{95}$ 0.46 | 97.20% | 6.5s |

each setup, we conducted 20 repeated experiments and recorded the success rates (success is defined as *block from verification*, unless the trainer did not modify the dataset). Additionally, we evaluated the usability metrics for each approach, namely the runtime and security risks. Regarding security risks, DI was not assessed because it does not alter the training process. For WM we tested the prediction accuracy of the backdoor samples it used. For DP, we evaluated the prediction accuracy of the data probe. The comparative results from Tab. 4 demonstrate that **only the Data-Probe approach successfully denied verification requests from attackers while exhibiting the lowest time expenditure and minimal security risks.**

**Practical modification.** We employed a representative and effective backdoor attack named Witches' Brew (Geiping et al., 2021). This attack alters 1% of the samples so that the trained model incorrectly classifies a targeted image, known as the trigger, as the wrong category. For instance, an automobile would be recognized as a deer. The results are displayed in Fig. 6, and the conclusions are consistent with those from the simulated experiments.

# 8 CONCLUSION

In this study, we highlight the importance of the Trustworthy Dataset Proof (TDP) in enhancing the veracity and integrity of training data for deep learning models. By introducing the novel Data Probe technique, this research successfully addresses the limitations of existing dataset provenance methods, which often falter in usability and integrity. The Data Probe, by leveraging subtle variations in model output distributions to verify the inclusion of specific training subsets, offers a model-agnostic and minimally invasive approach to dataset verification. Our extensive evaluations validate the effectiveness of our Data-Probe-based TDP framework, significantly advancing the pursuit of transparency and trustworthiness in training data usage.

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

## A  CHALLENGES FOR CURRENT TECHNIQUES

In this section, based on the formal definition of TDP in Sec. 3, we combine existing dataset trace-ability or copyright authentication technologies to propose some hypothetical solutions for TDP, and analyze the challenges involved.

**Case 1: Watermarking**

If we assume using watermarking technology (Adi et al., 2018; Tang et al., 2023) to achieve TDP, then the verifier can validate the occurrence of training behaviors based on two steps: watermark embedding and watermark detection. We use the subscript WM to represent the relevant functions of watermarking-based TDP.

In $\texttt{T-Train}_{\texttt{WM}}$, the trainer is required to embed watermarks into the model through training, and this operation can be denoted as $\mathbb{O}_{\texttt{WM}}$. $\mathbb{O}_{\texttt{WM}}$ allows the trainer to select some data samples from $\mathcal{D}$ as triggers $\mathbf{x}_t$ according to certain rules, which can make the trained model produce a specific pattern of output $\mathbb{P}$. We represent this characteristic as: $\mathbb{P} = \mathcal{M}(\mathbf{x}_t)$. Then the trainer sets $\mathcal{C} \leftarrow \mathbf{x}_t$ and submits it along with the trained model $\mathcal{M}$ to the verifier.

In the verification, the verifier could make the judgement:

$$\texttt{Verify}(\mathcal{D}, \mathcal{M}, \mathcal{C})_{\texttt{WM}} \leftarrow \mathbb{1}\left(\mathbb{P} = M(C) \wedge C \in D\right).$$

However, $\texttt{Verify}_{\texttt{WM}}$ cannot meet the defender's primary goal: fidelity **(G1)**, and we provide the simplest counterexample. The attacker can slightly manipulate the remaining data $\mathcal{D}_- = \mathcal{D} \setminus \mathbf{x}_t$ except for the triggers. It is noted that the current dataset $\mathcal{D}^* = \{\mathcal{D}_- \cup \mathbf{x}_t\} \neq \mathcal{D}$. Then the attacker execute $\texttt{T-Train}(\mathcal{D}^*)_{\texttt{WM}} \rightarrow \mathcal{M}^*, \mathcal{C}^*$. Here the $\mathcal{C}^* = \mathcal{C} = \mathbf{x}_t$ because the $\mathbf{x}_t$ remains unchanged. Since the watermark embedding mainly relies on $\mathbf{x}_t$, which means that the attacker can achieve $\mathbb{P} = \mathcal{M}^*(\mathbf{x}_t)$. Therefore, it is obvious that attackers can easily claim that the $\mathcal{M}^*$ are trained on $\mathcal{D}$ and verified by $\texttt{Verify}(\mathcal{D}, \mathcal{M}^*, \mathcal{C}^*)_{\texttt{WM}} = 1$.

**Case 2: Membership or Dataset Inference**

Membership Inference **(MI)** (Shokri et al., 2017; Salem et al., 2019) and Dataset Inference **(DI)** (Maini et al., 2021; Dziedzic et al., 2022) share similar approaches, both do not intervene during model training but directly analyze model outputs during the testing phase. We use the subscript MI and DI to represent the relevant functions of MI-based and DI-based TDP.

In training stage, $\texttt{T-Train}_{\texttt{MI}} = \texttt{T-Train}_{\texttt{DI}} = \mathbb{T}$ because $\mathbb{O}_{\texttt{MI}} = \mathbb{O}_{\texttt{DI}} = \varnothing$. During the verification, for the $\texttt{Verify}_{\texttt{MI}}$, the verifier may traverse all the data in $\mathcal{D}$ to determine if they belong to the training set. For the $\texttt{Verify}_{\texttt{DI}}$, the verifier could directly use the DI techniques to infer if $\mathcal{D}$ is the training set.

Nevertheless, both of them cannot satisfy the defender's primary goal: fidelity **(G1)** in principle. $\texttt{Verify}_{\texttt{MI}}$ evidently struggles to capture samples outside the claimed training set $\mathcal{D}$ because of the lack of information. On the other hand, DI can only ascertain the approximate data distribution of the training set. When the distributions of the training sets are similar, despite being unequal or even mutually exclusive, DI is highly likely to erroneously judge them as equivalent. In addition, DI is slightly disadvantaged in terms of efficiency **(G4)** because it needs to train a classifier for each verification request.

**Case 3: Proof of Training Data**

Assume we adopt the Proof of Training Data **(PoTD)** (Choi et al., 2024) to implement TDP, and denote the relevant functions with the subscript PT.

The $\texttt{T-Train}_{\texttt{PT}}$ requires the model trainer to record and provide all details during the training process, known as a training transcript $\mathbf{t}$, including training codes, various hyperparameters, and inter-mediate checkpoints. Then the trainer sets $\mathcal{C} \leftarrow \mathbf{t}$ and submits it to the verifier.

At the verification stage, the brute force solution of PoTD, that is, the verifier completely executing $\mathbf{t}$ to reproduce $\mathcal{M}$, can achieve the ideal TDP. However, this method is not acceptable in terms of computational cost, thus PoTD adopts some approximate verification methods to improve efficiency.

$\texttt{Verify}_{\texttt{PT}}$ still cannot meet the main goal of the defender: fidelity **(G1)**. As described in PoTD (Choi et al., 2024), this approximation for efficiency leads to the fact that "verifier will fails to catch spoofs

$\mathcal{D}$ if $\mathcal{D}$ only differs in a few data points." Furthermore, it has significant limitations in achieving defender's low-invasive goal **(G2)**. Because T-Train$_{\text{PT}}$ actually obtains white-box permission from the model trainer, an assumption that is sometimes impractical given that the training processes for many current models are considered commercial secrets.

## B  PROBE DETECTION METRIC

### B.1  PROBE SALIENCY AUC (**PSA**)

This metric utilizes probe scores to plot the ROC curve, further calculating the Area Under ROC curve (**AUC**) as the metric. We analyze this metric in detail from its priciple.

Assume here that we expect the probe scores $\mathbf{s}_p$ to be greater than non-probe scores $\mathbf{s}_{np}$ in distribution, as adopted in PP, for example. We do not focus on the actual score values but rather their relative magnitudes. Therefore, we could sort all scores from largest to smallest, sequentially selecting values as *hypothetical classification thresholds*. Samples with probe scores greater than the threshold are classified as probes, those lower as non-probes. Based on whether these samples are actually probes, we record the current predicted False Positive Rate (FPR) and True Positive Rate (TPR), hence obtaining an ROC curve.

Ideally, when all probe scores are greater than non-probe scores, the ROC curve will approach the top-left corner, with the corresponding Area Under the ROC Curve equaling 1. Conversely, when probe scores are nearly indistinguishable from non-probe scores, the process resembles a random guess, resulting in an ROC curve that goes from the bottom-left to the top-right corner, with an AUC close to 0.5. ***Hence, a larger AUC indicates that probe scores are more "separable" from non-probe scores***. We denote this as the Probe Saliency AUC (**PSA**) as an indicator of probe detection.

For PP and TP, we use the above approach to calculate PSA, which means predicting the saliency of data probe as 1 and non-probe as 0. However, for AP and UP, it is the opposite because we expect the scores of non-probe to be higher. Therefore, we predict probe as 0 and non-probe as 1 to calculate PSA.

### B.2  STATISTICAL TEST AND P-VALUE (**PV**)

Following several previous works (Maini et al., 2021), we perform a statistical t-test to measure whether there is a significant difference in the distributions of probe scores $\mathbf{s}_p$ and non-probe scores $\mathbf{s}_{np}$.

For PP and TP, the null hypothesis ($H_0$) is that the probe scores are less prominent compared to non-probe scores, which is opposite to our expectation. Assuming that $\mu_{\mathbf{s}_p}$ and $\mu_{\mathbf{s}_{np}}$ are the mean values of the $\mathbf{s}_p$ and $\mathbf{s}_{np}$, respectively. The $H_0$ and $H_1$ (alternate hypothesis) could be represented as:

$$H_0 : \mu_{\mathbf{s}_p} \leq \mu_{\mathbf{s}_{np}}; \quad H_1 : \mu_{\mathbf{s}_p} > \mu_{\mathbf{s}_{np}} \tag{1}$$

For AP and UP, the null hypothesis ($H_0$) is that the probe scores are more prominent compared to non-probe scores. We adopt the following hypothesis:

$$H_0 : \mu_{\mathbf{s}_p} \geq \mu_{\mathbf{s}_{np}}; \quad H_1 : \mu_{\mathbf{s}_p} < \mu_{\mathbf{s}_{np}} \tag{2}$$

The statistical t-test results in a p-value (**pV**), used as a metric to determine the success of the probe detection. Specifically, if the p-value is less than a certain level of significance, for example $0.1$, we reject the null hypothesis $H_0$, indicating that the probe was detected. Otherwise, we accept $H_0$ and consider that the probe was not detected.

## C  EVALUATION SETUP

**Datasets:** We adopt four datasets in our expreiments:

- **CIFAR-10** (Krizhevsky & Hinton, 2009): This dataset consists of 60,000 color images of 32x32 pixels, divided into 10 classes with 6,000 images per class. The dataset is split into

50,000 training images and 10,000 testing images. Classes include categories such as cars, birds, and cats.

- **SVHN** (Netzer et al., 2011): The Street View House Numbers (SVHN) dataset is derived from Google Street View, which features over 600,000 color images containing house numbers, formatted as 32x32 pixels.

- **CIFAR-100** (Krizhevsky & Hinton, 2009): Similar to CIFAR-10 but with 100 classes, this dataset includes 60,000 color images of 32x32 pixels, each class containing 500 training images and 100 testing images.

- **Tiny-ImageNet-200** (Le & Yang, 2015): It comprises 110,000 images across 200 classes, with each class represented by 500 training images, 50 validation images, and 50 test images. Each image is a 64x64 pixel color photograph. The dataset is a subset of the larger ImageNet collection, including a diverse array of categories ranging from various animal species to common everyday objects.

**Hyperparameters:** During training, the models are initialized randomly. The images in the dataset are uniformly resized to 224x224 and the pixel values are normalized to a range of -1 to 1 to comply with the model's input interface. To mitigate overfitting and ensure effective training of the model, random cropping and random horizontal flipping are employed during the training process. The Adam optimizer is adopted with a learning rate of 1e-3, and cross-entropy is employed as the loss function. Depending on the convergence of the model on various datasets, training is conducted for 10 to 15 epochs, and the model that performs best on the validation set is saved.

**Implementation details of TDP:** We sequentially present the implementation details of each critical operation within the TDP framework.

In the probe selection process, performing keyed-hash computations on datasets is computational-efficient. For example, running a keyed hash based on Md5 (Rivest, 1992) on the CI-FAR10 (Krizhevsky & Hinton, 2009) dataset on the computing platform with Intel Core i7-12700® takes an average time of only 1.73 seconds. To facilitate large-scale experiments without compromising the integrity of the framework's principles, we judiciously select fixed random seeds as a substitute for dataset hashing operations. We use the same random seed to select data probes, reflecting the scenario where the model trainer genuinely uses the dataset. Using different random seeds represents scenarios where a dishonest trainer initiates verification.

During probe implantation, the `WeightedRandomSampler` from the PyTorch is adopted for both PP and AP. Specifically, we assign a weight of 10 to the probes in PP, meaning they are ten times more likely to be selected during training compared to non-probes. For AP, the weight of the probes is set to 0.

When calculating probe scores, we employ cross-entropy for the loss-based score calculation, as cross-entropy is the most commonly used loss function in deep learning training tasks. Two metrics introduced in the Sec. 5: **PSA** and **p-value**, are adopted to assess whether probes can be effectively detected.

## D  ADAPTIVE ATTACKS VIA FORGE PROBE

Depending on the different types of probes, We have designed the following four targeted probe forging attacks in the evaluation, denoted with the prefix **F** for "Forged":

- **FPP**: The attacker fine-tunes $\mathcal{M}^*$ using $x_p$, attempting to enhance the prominence of $x_p$.

- **FAP**: The attacker "inversely" fine-tunes $\mathcal{M}^*$ using $x_p$, that is, employing a gradient ascent training method, attempting to diminish the prominence of $x_p$.

- **FUP**: The attacker uses $x_p$ and disrupts its label to fine-tune $\mathcal{M}^*$.

- **FTP**: The attacker uses $x_p$ and uniformly assigns them a random targeted label to fine-tune $\mathcal{M}^*$.

The principle for setting fine-tuning hyperparameters was to minimize the impact on the original performance of the model, for example, by using a very small learning rate of 2e-5 and training for 10 epochs.

---

**Algorithm 1:** TDP via Data Probe

---

1 **Function** T-Train$_{\text{DP}}(\mathcal{D})$:

   **Input:** Training dataset $\mathcal{D}$

   **Output:** Trained model $\mathcal{M}$, Certificate $\mathcal{C}$

2   Generate the key $\mathbf{k}$

3   Obtain indices $\mathbb{I}_p \leftarrow$ ProbeSelect($\mathcal{D}$, $\mathbf{k}$)

4   $\mathbf{x}_p \leftarrow \mathcal{D}[\mathbb{I}_p]$

5   Operate the probe with $\mathbb{O}_{\text{DP}}(\mathbf{x}_p)$

6   $\mathcal{M} \leftarrow \mathbb{T}(\mathcal{D}), \mathcal{C} \leftarrow \mathbf{k}$

7   Return: $\mathcal{M}, \mathcal{C}$

8 **Function** Verify$_{\text{DP}}(\mathcal{D}, \mathcal{M}, \mathcal{C})$:

   **Input:** Claimed training dataset $\mathcal{D}$, Trained model $\mathcal{M}$, Certificate $\mathcal{C}$

   **Output:** Verification result $\{0, 1\}$

9   Obtain indices $\mathbb{I}_p \leftarrow$ ProbeSelect($\mathcal{D}$, $\mathcal{C}$)

10   $\mathbf{x}_p \leftarrow \mathcal{D}[\mathbb{I}_p], \mathbf{x}_{np} = \{\mathcal{D} \smallsetminus \mathbf{x}_p\}$

11   $\mathbf{s}_p \leftarrow$ ProbeScore($\mathcal{M}$, $\mathbf{x}_p$)

12   $\mathbf{s}_{np} \leftarrow$ ProbeScore($\mathcal{M}$, $\mathbf{x}_{np}$)

13   Return $\mathbb{1}(\mathbf{s}_p \neq \mathbf{s}_{np})$

---

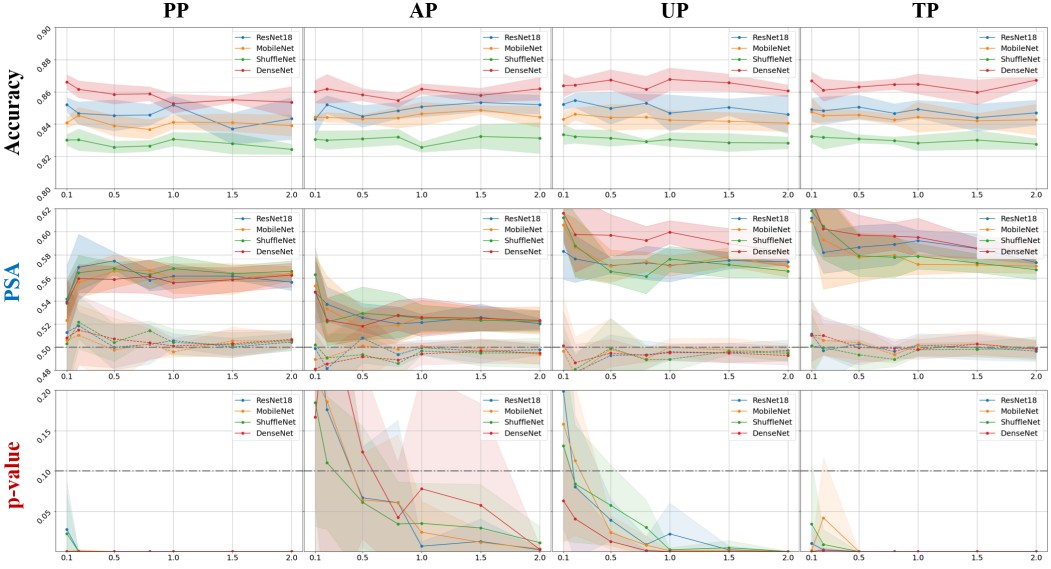

Figure 7: **Evaluations of the Impact of Probe Implant Quantity on TDP Performance.** The horizontal axis represents the number of probes as 0.1% ~ 2% of the total training dataset. Additionally, we use a gray dashed line to mark two critical reference values: PSA = 0.5 and p-value = 0.1. A PSA significantly above 0.5 and a p-value below 0.1 indicate successful detection of data probes.

| Description | Sample | Original Trigger Score | Watermark Verify Result | Watermark Trigger Score | Dataset Inference Verify Result | Dataset Inference Trigger Score | Data Probe (Ours) Verify Result | Data Probe (Ours) Trigger Score | HASH (train) | HASH (test) |
|---|---|---|---|---|---|---|---|---|---|---|
| Automobile ↓ Deer | | 99.82 Deer | PASSED 1.000 | 84.24 Deer | PASSED 2.3×10⁻⁷ | 99.82 Deer | BLOCKED 0.85 | 98.06 Deer | 90db8061ae4 0dcbd47f31bf 17e173651 | 9504b703430 662aad7aa7c 4a9f0333b2 |
| Frog ↓ Truck | | 99.42 Truck | BLOCKED 0.487 | 96.67 Frog | PASSED 2.4×10⁻⁷ | 99.42 Truck | BLOCKED 0.88 | 70.99 Truck | f91974d8a4c0 7fe8f0b82986 317664ff | 5aa11da5aee3 d6e48425ac3 1d8d5dc82 |
| Deer ↓ Bird | | 99.53 Bird | PASSED 0.743 | 94.46 Bird | PASSED 2.5×10⁻⁷ | 99.53 Bird | BLOCKED 0.72 | 99.98 Bird | eaafcb63f846 3d685b842df eb2829a37 | e32c6418fd5e ee870efa547 96d47ccbe |
| Dog ↓ Horse | | 91.94 Horse | PASSED 0.907 | 99.91 Horse | PASSED 2.8×10⁻⁷ | 91.94 Horse | BLOCKED 0.59 | 98.42 Horse | 288168d220ff 9384b373705 f27a7c48f | 7cb416bd1528 8c76f5411819 b0d8f85c |
| Horse ↓ Cat | | 76.79 Cat | BLOCKED 0.053 | 80.92 Cat | PASSED 2.2×10⁻⁷ | 76.79 Cat | BLOCKED 0.74 | 98.89 Cat | afb6d130afb7 f640f143179f 8a3f21ef | a6e6c6c6e13c 3b53d54e6de 6b4af9b9d |

Figure 8: **Case Studies for Comparison with Existing Techniques.** The scores for each validation are presented in GRAY, with watermarking representing the predicted probabilities, and DI and Data Probe indicating the p-values. HASH(train) shows the hash value computed on the tampered dataset during probe implantation, which differs from the HASH(test) obtained on the declared untampered dataset (CIFAR-10) during testing.