# OpenReview forum: "Trustworthy Dataset Proof: Certifying the Authentic Use of Dataset in Training Models for Enhanced Trust"
_ICLR.cc/2025/Conference — ICLR 2025 Conference Withdrawn Submission_

### Official Review · Reviewer_ty3A · 2024-11-02

**Soundness:** 2
**Presentation:** 2
**Contribution:** 2
**Rating:** 5
**Confidence:** 4

**Summary:**

This study presents Trustworthy Dataset Proof (TDP), a new approach to verifying the authenticity of training data in deep learning, addressing a gap left by existing dataset provenance methods that focus on ownership rather than trust enhancement. Traditional methods face challenges like high operational demands and ineffective dataset verification. To overcome these limitations, the authors propose Data Probe, a model-agnostic technique that leverages subtle variations in model output distributions to detect the presence of specific subsets of training data, differing from watermarking methods. This approach enhances usability by reducing training intervention and ensures dataset integrity, allowing detection only when the full claimed dataset is used. Extensive evaluations demonstrate the efficacy of the Data Probe-based TDP framework, advancing transparency and trust in training data utilization in deep learning.

**Strengths:**

1.	The proposed problem of trust dataset proof is interesting and important. This work provides an effective solution letting model trainers claim authentic usage of certain datasets.

2.	The proposed methods don’t rely on cryptography techniques, thus being lightweight and efficient.

3.	The paper is well-organized and easy for readers to follow.

**Weaknesses:**

1.	The verification protocol is unclear in terms of presentation. For example, in probe selection, there is a need to generate or specify a key to select the data probe, who is responsible for generating this key, the paper sometimes adopts ‘trainer’ and sometimes adopts ‘users’, which is very confusing. Figure 3 cannot depict the protocol clearly as well.

2.	The definition of harmlessness is too narrow, which is much wider than model performance compromising. Actually, the proposed four types of data probes all introduce harm to the model. For example, Prominent Probe leverages higher overfitting that increases the vulnerability of membership inference attacks. Absence Probe does hurt the performance of data probes and similar data. Needless to say, the Untargeted and Targeted Probe. In terms of general harmless dataset provenance, I believe this work [1] provides a better solution.

3.	According to the experiment results, different types of data probes perform distinctively in different metrics. However, this paper does not provide a solution that can unify different types of data probes to achieve a comprehensively better type. Without this, in practical scenarios, how does the user know which type to choose for specific cases?

4.	In lines 362-363, the paper mentions that the sample-level scores are aggregated to form a new metric, but I cannot find the detailed aggregation approach. Then it is unclear how and where PSA in Table 2 originates from.

5.	As for the adaptive attack, I disagree that replicating and embedding the probe corresponding to D after training would be adopted by the attacker. The more potential way is replicating and embedding the probe during the training. Besides, the experiment setup is also unclear, like how is the modified dataset D* made and how large the difference between D and D*?

6.	According to my understanding of AUC, I don’t think a score exceeds 50 but smaller than 60 is sufficient for distinguishing two distributions.

7.	Figure 4 needs improvement, like there is no legend for the dashed curve and the y-axis of the third subfigure can use a log scale.

[1] Guo, Junfeng, et al. "Domain watermark: Effective and harmless dataset copyright protection is closed at hand." Advances in Neural Information Processing Systems 36 (2023).

**Questions:**

See the weaknesses above.

---

> ### Author Response · Authors · 2024-11-23
> **Reply to Reviewer ty3A (1/2)**
>
> Many thanks for your detailed review and insightful comments.  We will carefully consider your recommendations to improve the manuscript and respond to your questions in detail, hoping to address your concerns.
>
>  # $\textcolor{red}{{\rm Q.1}}$ Secret Key and User
>
> **Who is the user ?**
>
> In the main text, "user" refers to the "model trainer" or, in the case of a dishonest trainer, the "attacker." We provide a detailed analysis of this in the General Response $\textcolor{red}{{\rm GR.1.1}}$.
>
> **Who generate the key ?**
>
> In $\textcolor{red}{{\rm GR.5}}$, we provide a detailed analysis of various scenarios regarding who is responsible for generating the key. Besides, in $\textcolor{red}{{\rm GR.3}}$ we elaborate on how to implement the whole verificaition framework.
>
> **Planned Revisions**
>
> We sincerely thank you for your valuable comments on this part. We will standardize the terminology throughout the manuscript by replacing "user" with "trainer". Considering the page constraints in the main text, we will include the implementation details in the appendix to better support the verification protocol described in the main text.
>
>  # $\textcolor{red}{{\rm Q.2}}$ Definition of Harmlessness
>
> ndeed, as you pointed out, PP may make probes more susceptible to MIA attacks. However, we believe that in the TDP scenario, the focus is on the dataset's trustworthiness rather than its privacy. Since the model trainer has already publicly declared the use of dataset $D$, o gain trust, users can clearly identify the members of $D$.
>
> Therefore, we consider the focus on performance compromising, rather than privacy, to be reasonable for the definition of harmlessness in this work. We provide a detailed explanation of the dataset's nature in $\textcolor{red}{{\rm GR.2}}$, emphasizing the key distinctions between TDP and dataset provenance tasks. Additionally, we included an example in $\textcolor{red}{{\rm GR.1.2}}$ to further illustrate these differences. We hope this helps address your concerns.
>
> In terms of performance, exactly, AP slightly hurts the performance of probes. Similarly, UP and TP affect the accuracy of the probes themselves. However, our primary focus is on the performance of the test set, as in deployment scenarios, models are most likely to encounter non-training data. Generalization performance is thus more critical.
>
> The extensive experimental results in $\textcolor{green}{{\rm Table 2}}$ demonstrate that the impact on generalization is negligible, mainly due to the small proportion of probes in the dataset. Therefore, we believe that the harmlessness of AP, UP, and TP is within an acceptable range.
>
> Certainly, we deeply appreciate your valuable opinion. Investigating whether applying overfitting to certain samples might introduce additional security risks in the TDP scenario is indeed a meaningful topic for further exploration. Although it may go beyond the scope of the current work, we plan to delve into this aspect in future studies.
>
>
>  # $\textcolor{red}{{\rm Q.3}}$ Probe Type Selection Strategy
>
> **Why different probe?**
>
> Since TDP is a completely new task with no available baselines, this work, inspired by MIA and poisoning attacks, designs four types of probes primarily to serve as baselines for each other. The goal is to provide TDP framework users with a reference for selecting probes by comparing their performance. Similarly, we designed four scoring methods for probes and demonstrated through experiments that Mentr achieves the best performance.
>
> **Which probe to be used?**
>
> From the comparison of the results in $\textcolor{green}{{\rm Table 2}}$ and $\textcolor{green}{{\rm Table 3}}$, it is evident that **PP and AP exhibit better generalizability**. As dataset complexity increases (SVHN → CIFAR-10 → CIFAR-100 → TinyImageNet-200), the overfitting effect on probes becomes significantly stronger, demonstrating their applicability to more practical datasets.
>
> In contrast, UP and TP may face the risk of failure when dealing with complex datasets, as their effectiveness is more heavily influenced by the model's ability to fit the dataset. However, in terms of robustness, **UP and TP offer better security** compared to AP and PP.
>
> **Planned Revisions**
>
> We will enhance the discussion on probe type selection in the manuscript.  Additionally, combining the strengths of the four types of probes to achieve a comprehensively better type is certainly a valuable research direction. While it may slightly exceed the scope of the current paper, we will consider exploring it in future work.

---

> > ### Author Response · Authors · 2024-11-23
> > **Reviewer ty3A (2/2)**
> >
> > # $\textcolor{red}{{\rm Q.4}}$ Concerns of PSA
> >
> > Due to space constraints in the main text, we have included the calculation details of PSA in  $\textcolor{green}{{\rm Appendix~B.1}}$.
> >
> > Specifically, the calculation of PSA is based on the principle of the AUC curve, where sample-level scores are ranked, and FPR (False Positive Rate) and TPR (True Positive Rate) are computed at different thresholds.
> >
> > Regarding the choice of AUC thresholds, unlike traditional classification tasks where an AUC above 60 is required for robust classification, in this task, it is sufficient to distinguish whether the probe's score response is slightly higher than non-probe scores. When no correct probe is identified, the scores for probes $s_p$ and non-probes $s_{np}$ should exhibit very similar distributions, meaning they will be evenly distributed during ranking.  This results in PSA stabilizing around 0.5 (as seen in all gray scores in $\textcolor{green}{{\rm Table 2}}$). Therefore, if PSA significantly exceeds 0.5, such as reaching around 0.52, it is already sufficient to distinguish between the two distributions.
> >
> > While p-values (pV) can largely indicate whether these two distributions differ significantly, given the lack of more established metrics for the TDP task, we propose PSA as a complementary metric. Moreover, extensive experiments show that **PSA and p-values tend to exhibit consistent trends** (e.g., when PSA > 0.52, p-values are often far below 0.1, confirming distributional differences).  This demonstrates PSA's usability and reliability.
> >
> > # $\textcolor{red}{{\rm Q.5}}$ Concerns of Adaptive Attack
> >
> > **Significance of the proposed adaptive attack**
> >
> > Indeed, as you pointed out, we provide a detailed analysis in $\textcolor{red}{{\rm GR.4}}$ of the TDP system's security under different threat scenarios. When the attacker knows all verification details, they could potentially undermine the verification system by directly simulating and manipulating the data probe embedding process. However, we consider this assumption about the attacker's capability **overly strong** and, to some extent, contradictory to our assumption that the trainer cannot manipulate the data probe embedding process. In $\textcolor{red}{{\rm GR.3}}$, we also explained the rationale and feasibility of preventing the attacker from gaining full knowledge of the system.
> >
> > Therefore, we plan to revise the description of the Forge Probe adaptive attack. It targets the security setting where the attacker has partial knowledge of the verification details and needs to infer the remaining information (e.g., the hash function). Forge Probe serves as a **rapid strategy** for guessing probe positions under such circumstances. We have also provided a detailed analysis and discussion of this in $\textcolor{red}{{\rm GR.4}}$ and strongly encourage you to refer to that section for further clarification.
> >
> > **Experimental setting of the adaptive attack**
> >
> > In the adaptive attack experiment, we assume that the difference between $D$ and $D^*$ is minimal, such as only a single sample being modified.
> >
> > To facilitate large-scale experiments without compromising the integrity of the framework’s principles, in the adaptive attack evaluation, we directly control the random seeds as a substitute for dataset modification and hashing operations.
> >
> > Specifically, even minimal differences in the dataset can result in a change of the seed (as illustrated in $\textcolor{green}{{\rm Figure~6}}$ of the case study). During training, we use seed $s_1$ (simulating the scenario of training with $D^*$), and then attempt to forge the probe corresponding to seed  $s_2$ (representing training with $D$)
> >
> > We will adjust and better elaborate on this experimental setup in the main text.

---

> > > ### Comment · Reviewer_ty3A · 2024-11-25
> > >
> > > Thanks for the detailed response. After careful reading, I still have the following concerns.
> > >
> > > 1. Although the author(s) acknowledged the harmlessness in terms of the generalization of testing data, such harm does expose the vulnerability to various attacks not limited to MIA. The comparison between the proposed work and [1] is lacking.
> > >
> > > 2. I partially disagree with some claims in GR.2. For example, one of the most representative application scenarios of dataset provenance is data copyright protection, where the dataset is public and dataset provenance is used to validate whether a target model is trained on the protected dataset. I don't think there are very significant differences between the TDP and dataset provenance in terms of problem setups and objectives, and using such differences to prevent the comparison with existing state-of-the-art dataset provenance is unreasonable. I agree with the difference in terms of who should use TDP and dataset provenance. However, comparison experiments are also needed, as dataset provenance can be used in TDP scenarios as well.
> > >
> > > 3. I don't think combining different types of probes exceeds the scope of the current paper. If it needs extensive effort and cannot be done in the current submission, I suggest finishing it and seeking other venues instead.
> > >
> > > 4. I still have no idea why AUC here does not need to exceed 60 to distinguish different distributions. Is it possible to visualize the sample score ranking?

---

> > > > ### Author Response · Authors · 2024-11-27
> > > > **Thanks for Replying**
> > > >
> > > > Thank you for your reply! We will carefully consider your feedback and further improve our work.

---

### Official Review · Reviewer_stcP · 2024-11-03

**Soundness:** 2
**Presentation:** 3
**Contribution:** 2
**Rating:** 5
**Confidence:** 3

**Summary:**

The paper introduces a novel framework called Trustworthy Dataset Proof (TDP), aimed at ensuring the authenticity and integrity of training datasets. Unlike traditional watermarking, Data Probe uses subtle variations in the output distributions of models trained on specific data, allowing for the detection of a small, representative subset of the training data. Also, this proposed approach is model-agnostic. Experimental results demonstrates the effectiveness of this approach, showing that the data probe-based TDP framework is both practical and reliable in verifying dataset integrity.

**Strengths:**

+) The topic of trustworthy proof in deep learning training is timely and interesting

+) The paper is well organized and easy to follow

+) The proposed method is technically reasonable and sound

**Weaknesses:**

-) The motivation and threat model is not clear to me

-) Some assumptions in the problem setting are not very reasonable

-) There lacks a theoretical analysis (or proof) how well the proposed method will work

**Questions:**

a) This paper addresses a gap in the field by introducing the novel problem of verifying the authentic usage of datasets in deep learning. The topic is interesting as the community pushes toward responsible AI practices amid growing concerns about data misuse and authenticity. By focusing on dataset integrity through Trustworthy Dataset Proof (TDP), the authors make a timely contribution to the field.

b) My main concern is the motivation and threat model is not clear to me. The paper proposed a method to verify the training was performed on 'a credit dataset D' such that 'the model trainer can gain trust from the users'. This claim seems to be not convincing to me. Why a user can have extra trust if the model is trained a specific dataset? What is the goal of the model user? It would be better to show some real examples to help clarify it.

c) Some assumptions used in the paper need justification. For example, the goal of the paper is to verify if the entire dataset was authentically used to train a model. However, in practice, a dataset is often combined with other datasets to jointly train a model - this will change the distribution and the proposed method may not work. Also, trainer may apply on-the-fly augmentations/filters which may affect the training data distribution. It would be better to discuss how these would impact the proposed method.

d) While the evaluation results show that the proposed method can work well, there lacks a theoretical analysis (or proof) on the effectiveness of the proposed method. The evaluated datasets are small datasets, therefore there might be a gap when we use it in practice. A theoretical analysis will help understand the gap and better justify the proposed method.

e) Meanwhile, the adaptive attack results seems not very convincing to me. In this setting, the attacker has fully knowledge of the verification process and the exact data probe to be used for verification, so why the attacker cannot simply overfit the desired model output of the probe data?

---

> ### Author Response · Authors · 2024-11-23
> **Reply to Reviewer stcP (1/2)**
>
> We deeply appreciate your thorough review and constructive feedback, as well as your kind acknowledgment of our work. We will carefully consider your recommendations to improve the manuscript and respond to your questions in detail in the following sections.
>
> # $\textcolor{red}{{\rm Q.1}}$ Motivation of the Work
>
> > My main concern is the motivation and threat model is not clear to me. The paper proposed a method to verify the training was performed on **'a credit dataset D'** such that 'the model trainer can gain trust from the users'. This claim seems to be not convincing to me. Why a user can have extra trust if the model is trained a specific dataset? What is the goal of the model user? It would be better to show some real examples to help clarify it.
>
> In Genral Response $\textcolor{red}{{\rm GR.1}}$, we provide a detailed introduction to the potential application scenarios of TDP, illustrated with an example based on the real-world model publishing platform Hugging Face.
>
> We further build on this example to explain *"why a user can have extra trust if the model is trained on a specific dataset."*. More precisely, in the TDP problem, the focus of both the verifier and the model user is **not on which dataset is used** but rather on **how the dataset is used**. Therefore, the statement is more accurately expressed as: "a user can have extra trust if the model is trained on a specific dataset **without modification.**"
>
> Taking CIFAR-10 as an example, as mentioned in $\textcolor{red}{{\rm GR.1}}$, our case study ($\textcolor{green}{{\rm Figure~6}}$ in the manuscript) experimentally demonstrates that an attacker can modify a small portion of CIFAR-10 samples to embed backdoors while still claiming that the model is trained on pristine CIFAR-10. Such a model can bypass existing dataset provenance-based techniques.  If the model user trusts the attacker’s claim and deploys the model, it clearly introduces significant security risks.
>
> Conversely, if there is a mechanism—such as the TDP task proposed in this paper—that can distinguish whether CIFAR-10 was modified during training, users would naturally prefer the model without modifications. This provides the model with extra trust.
>
> In summary, the TDP framework we propose is specifically designed to verify that the training was performed on "a credit dataset $D$" without modification. Here, $D$ can be CIFAR-10 or any other dataset. We also provide a detailed discussion on the "credit" property of $D$ in $\textcolor{red}{{\rm GR.2}}$.
>
> # $\textcolor{red}{{\rm Q.2}}$ More Complex Scenarios
>
> **Planned Revisions**
>
> Thank you very much for the insightful suggestion. We indeed recognize that, in real-world application scenarios, there will be more complex issues that do not fully align with our foundational assumptions. Due to the page constraints of the manuscript, we plan to expand this discussion in the revised version, either in the discussion section or the appendix.
>
> **Extension to Multiple Datasets**
>
>
> We consider this can be broadly divided into two situations:
>
> 1. Using two similar datasets, merging them as a single unit for model training.
> 2. Pretraining on a large, general-purpose dataset (e.g., using ImageNet to pretrain the backbone) followed by fine-tuning on a smaller domain-specific dataset.
>
> Our approach can be conveniently extended to accommodate the first scenario. For example, we can treat two datasets, $D_1$ and $D_2$ , as a single unit and perform a single data probe implantation (by concating them). During verification, the trainer simply declares to the verifier that the model was trained using $D_1 + D_2$.
>
> For the second situation, our approach may have certain limitations. For instance, if probes are first embedded when trained with $D_1$ and then additional probes are embedded during fine-tuning with $D_2$, the distribution may indeed change, potentially affecting the stability of the $D_1$'s probes. We acknowledge that this is a meaningful topic for further exploration and plan to investigate it in future work.
>
> **Adapting to Data Augmentation**
>
> Our algorithm can be adapted to support basic data augmentation. Specifically, we compute the hash on the complete dataset without any augmentation, select the data probes, and apply specific treatments (e.g., increasing the selection probability for PP). During training, data augmentation is allowed as part of the trainning process. The implementation of this adaptation can also be understood in conjunction with the trusted-training wrapper described in $\textcolor{red}{{\rm GR.3}}$.
>
> In fact, all experimental results in this study were obtained with data augmentation techniques such as horizontal flipping and random cropping during training (as detailed in $\textcolor{green}{{\rm Appendix C}}$). When employing more extensive augmentation or filtering techniques, the situation may become more complex and warrants further experimental exploration. We plan to consider this as future work.

---

> > ### Author Response · Authors · 2024-11-23
> > **Reply to Reviewer stcP (2/2)**
> >
> > # $\textcolor{red}{{\rm Q.3}}$ Concerns of Adaptive Attack
> >
> > Indeed, as you pointed out, we provide a detailed analysis in $\textcolor{red}{{\rm GR.4}}$ of the TDP system's security under different threat scenarios. When the attacker knows all verification details, they could potentially undermine the verification system by directly simulating and manipulating the data probe embedding process. However, we consider this assumption about the attacker's capability **overly strong** and, to some extent, contradictory to our assumption that the trainer cannot manipulate the data probe embedding process. In $\textcolor{red}{{\rm GR.3}}$, we also explained the rationale and feasibility of preventing the attacker from gaining full knowledge of the system.
> >
> > Therefore, we plan to revise the description of the Forge Probe adaptive attack. It targets the security setting where the attacker has partial knowledge of the verification details and needs to infer the remaining information (e.g., the hash function). Forge Probe serves as a **rapid strategy** for guessing probe positions under such circumstances. We have provided a detailed analysis and discussion of this in $\textcolor{red}{{\rm GR.4}}$ and strongly encourage you to refer to that section for further clarification.
> >
> > # $\textcolor{red}{{\rm Q.4}}$ Theoretical Analysis
> >
> > We greatly appreciate your suggestion, and enhancing the theoretical analysis and proof is indeed one of our key focus areas for improving this work.
> >
> > We also acknowledge that theoretically proving the effectiveness of data probes is not a trivial task. Taking PP as an example, its effectiveness relies on the phenomenon of model overfitting. In the most relevant theoretical work, DI [1] provides an analysis of model overfitting and validation effectiveness only for linear models. For more complex, nonlinear neural networks, the analysis was conducted experimentally rather than theoretically.
> >
> > Additionally, the datasets evaluated exhibit increasing complexity (SVHN → CIFAR-10 → CIFAR-100 → TinyImageNet-200). TinyImageNet, as a representative subset of the large-scale ImageNet dataset, provides some degree of support for assessing its practical effectiveness.
> >
> > Of course, we will continue to work on improving the theoretical analysis. While we anticipate that it may be challenging to complete this during the current discussion phase, we will strive to address it in the revised version or in future extended work. Thank you again for your valuable suggestion.
> >
> > [1] Dataset inference: Ownership resolution in machine learning.

---

### Official Review · Reviewer_tebf · 2024-11-03

**Soundness:** 2
**Presentation:** 3
**Contribution:** 2
**Rating:** 5
**Confidence:** 3

**Summary:**

This paper focuses on a watermarking approach for certifying dataset use in model training. It targets the research gap of usability and integrity of proposed solutions to the Trustworthy Data Proof problem. The paper’s goals include fidelity of the watermarking verification approach, low-invasiveness on the model training process, harmlessness (ie. minimizing the model’s performance degradation), and efficiency of the verification process.
The authors propose Data Probe, an approach conceptually similar to a weakened backdoor, where the probe input samples behave like special members of the dataset. However, Data Probe requires only a slight difference in model prediction confidence between the probe versus non-probe inputs  in order for the dataset to be successfully verified, as opposed to backdoors which trigger a specific output. The success of their approach relies on coupling of the dataset integrity verification with the data probe selection strategy, utilizing the uniqueness of hash functions. In particular, a user-specific keyed-hash is performed on the complete.
The authors propose and analyse the performance of 4 different types of data probe, as well as 4 different possible saliency scoring methods. Experiments are carried out to answer 4 RQs -  whether various types of data probes can effectively verify the integrity of the dataset, how many probes need to be implanted during training, which of the different probe score calculation strategies are most effective for detection, and how robust the verification mechanism is against adaptive attacks. Experiments show promise for their proposed Data-Probe-based solution to the TDP problem.

**Strengths:**

TDP is an interesting and underexplored problem, that is well motivated in the paper.
Overall the paper is well written and reasonably easy to understand, and the threat model seems reasonable, with the defender only requiring black-box access to the model to be verified.
Overall results look promising on a decent range of experiments, including adaptive attack.

**Weaknesses:**

* The main weakness is that overall, while results for adaptive attack and case studies look promising, the experiments are only using Cifar-10 – which has only 10 classes of relatively low-resolution images (32x32 pixels), and each analysis is only performed on one NN architecture. This is not comprehensive enough to establish that the performance of Data Probe generalizes to diverse datasets and model architectures.
* a missed related work: “DeepTaster: Adversarial Perturbation-Based Fingerprinting to Identify Proprietary Dataset Use in Deep Neural Networks” by Park et. al. (ACSAC ’23). This work identifies dataset use in model training with no invasiveness or harm in the approach. It may also be possible to compare overall accuracy of Data Probe with their model.
* It is not stated what scores are shown in gray in Table 4.
* In the adaptive attack experiment, the difference between the data D Vs D*  is not clear.
* In the case study – duplicating data to create extra data is not very compelling. It would be more interesting to see something like the SVHN dataset augmented with some MNIST data.

Typos/etc:
- in the abstract: “data-drobe-based".
- Please check the definition of Conf(M,x) on page 7. What is the maximum taken over?
- Table 4 caption: “socres” should be “scores”

**Questions:**

To improve the paper, please address the following:
* Expand the adaptive attack and case study experiments to include all the datasets and NN architectures covered in the Table 2 results (SVHN, CIFAR-100, Tiny-Image-Net-200, ResNet, MobileNet, DenseNet, ShuffleNet).
* Add a clear explanation of what the gray scores represent in the Table 4 caption or in the accompanying text discussing Table 4. This would help improve the clarity of their results presentation.
* Add a discussion of “DeepTaster" by Park et. al. (ACSAC ’23) to the related work and compare the fidelity of their approach (if appropriate).
* Provide a more detailed explanation of how D* differs from D in the adaptive attack scenario. Eg. include a specific example to clarify this important aspect of the experimental setup.

---

> ### Author Response · Authors · 2024-11-23
> **Reply to Reviewer tebf**
>
> Thank you very much for your thorough review and valuable feedback, particularly your suggestions regarding experimental evaluation and related work. We will carefully consider your advice to improve our work. In the following sections, we provide detailed responses to your questions in the hope of addressing your concerns.
>
> # $\textcolor{red}{{\rm Q.1}}$ Expand the evaluation
>
> Thank you for your suggestion. We will take it into account to improve our adaptive attack and case studies experiments. Due to time constraints, we may not be able to complete and organize all the results during the rebuttal period. We sincerely apologize for this in advance, but we would try our best include these experiments in the final revised version.
>
> # $\textcolor{red}{{\rm Q.2}}$ Comparison To DeepTaster
>
> **Principle of DeepTaster: similar to DI**
>
> We analyzed the principle of DeepTaster. In terms of classification, it employs a DNN fingerprinting technique different from watermarking. Broadly speaking, it utilizes specific techniques to compute the response differences of a model to certain samples, thereby extracting the model's fingerprint, and then train an additional classifier to make determinations. This principle is almost identical to that of **DI**.
>
> Therefore, we believe that DeepTaster shares **similar characteristics with DI**: it is non-invasive but requires a high verification overhead (as a classifier needs to be trained individually for each verification). Additionally, it struggles to distinguish between two similar data distributions (e.g., $D$ and $D^*$  with minimal modifications).
>
> **Planned Revisions**
>
> We will include an introduction and discussion of DeepTaster in the related work section.
>
>
> # $\textcolor{red}{{\rm Q.3}}$ Other Questions
>
> > - It is not stated what scores are shown in gray in Table 4.
> > - Add a clear explanation of what the gray scores represent in the Table 4 caption or in the accompanying text discussing Table 4. This would help improve the clarity of their results presentation.
>
> The gray scores in $\textcolor{green}{{\rm Table4}}$ represents the scores for each validation, with watermarking representing the predicted probabilities, and DI and Data Probe indicating the p-values.
>
> As it conveys the same meaning as $\textcolor{green}{{\rm Table 6}}$, we directly referenced $\textcolor{green}{{\rm Table 6}}$'s explanation in $\textcolor{green}{{\rm Table~4}}$ due to space constraints. We will work on refining and improving the explanation in this caption.
>
> ---
>
> > - In the adaptive attack experiment, the difference between the data D Vs D* is not clear.
> > - Provide a more detailed explanation of how D* differs from D in the adaptive attack scenario. Eg. include a specific example to clarify this important aspect of the experimental setup.
>
> In the adaptive attack experiment, we assume that the difference between $D$ and $D^*$ is minimal, such as only a single sample being modified.
>
> To facilitate large-scale experiments without compromising the integrity of the framework’s principles, in the adaptive attack evaluation, we directly control the random seeds as a substitute for dataset modification and hashing operations.
>
> Specifically, even minimal differences in the dataset can result in a change of the seed (as illustrated in $\textcolor{green}{{\rm Figure~6}}$ of the case study). During training, we use seed $s_1$ (simulating the scenario of training with $D^*$), and then attempt to forge the probe corresponding to seed  $s_2$ (representing training with $D$)
>
>  We will adjust and better elaborate on this experimental setup in the main text.
>
> ---
>
> > Typos/etc:
> > - in the abstract: “data-drobe-based".
> > - Please check the definition of Conf(M,x) on page 7. What is the maximum taken over?
> > - Table 4 caption: “socres” should be “scores”
>
> Due to space constraints, we have provided an explanation of this part in $\textcolor{green}{{\rm Appendix.C}}$. Similarly, we will work on refining and improving this explanation in the main text.
>
> Thank you for your suggessions. We will address these typos in the revised version and make every effort to correct any other existing typos.
>
> Upon our review, the definition of Conf(M,x) is correct, representing the maximum taken over the logits of the model's output. The design of this function is motivated by [1]
>
> [1] Ml-leaks: Model and data independent membership inference attacks and defenses on machine learning models.

---

> > ### Comment · Reviewer_tebf · 2024-11-25
> >
> > Thank you to the authors for the detailed response, particularly the additional information on the threat model and adaptive attack. However, without the results of the extended evaluation and a finalized writeup of the threat model details, I will maintain my current score. Additionally, I agree with the other reviewers that dataset provenance could be applied in TDP, making a comparison with state-of-the-art dataset provenance methods necessary.

---

> > > ### Author Response · Authors · 2024-11-27
> > > **Thanks for Replying**
> > >
> > > Thank you for your reply! We will carefully consider your feedback and further improve our work.

---

### Official Review · Reviewer_5EGH · 2024-11-04

**Soundness:** 1
**Presentation:** 2
**Contribution:** 3
**Rating:** 3
**Confidence:** 4

**Summary:**

This paper introduces the Trustworthy Dataset Proof (TDP) problem, which aims to verify the authenticity of training datasets used to train machine learning models. TDP provides a mechanism to certify that a claimed dataset was used fully, without additional tampering, during model training, thus enhancing trust and transparency. The paper proposes a novel “Data Probe” method, which embeds subtle markers in the dataset and uses model output distribution to confirm dataset integrity with minimal intervention in the training process. The authors demonstrate TDP’s performance on several vision datasets, showing that it can detect when training data is tampered with or incomplete.

**Strengths:**

**Definition of a New Dataset Verification Problem**

The paper identifies and defines the novel problem of Trustworthy Dataset Proof (TDP), which seeks to verify the integrity of datasets used in model training. Unlike existing approaches focused on dataset ownership or provenance, TDP aims specifically to ensure that the declared dataset was used in its entirety.

**Novel Approach to Address the Problem**

The authors propose a novel solution to the TDP problem by introducing the concept of “Data Probes.” This approach marks subsets of the dataset in a way that allows verification through model outputs with minimal impact and modifications to the training process. This probe-based strategy leverages distributional output patterns to detect incomplete or tampered datasets, and only requires black-box access to the model.

**Efficiency and Low Overhead**

The TDP framework is designed to be computationally efficient, requiring only minimal overhead compared to regular training. This is beneficial compared to approaches like Proof of Training Data, which incur significant overhead during training.

**Weaknesses:**

**Inconsistent Threat Model Assumptions**

The main limitation is that the threat model assumes a dishonest trainer (L161) who attempts to modify the dataset and avoid detection. At the same time, the threat model assumes that the same dishonest attacker voluntarily sticks to the exact verification protocol specified by the verifier (defender), which cannot be verified (L192).

In practical terms, this means it is easy for the attacker to bypass the proposed verification approach. Instead of computing the hash over the modified dataset D*, the attacker computes the hash over the unmodified dataset D, and thus always reaches the goal of succeeding the verification.

**The Dataset has to be Public**

The dataset has to be entirely available to verify the trained model. Since the data contains a significant part of the value, the trainer may be reluctant to publish it. Licenses and intellectual property may also work against this.

**Limited Evaluation**

The evaluation is limited to small image classification datasets: CIFAR-10, SVHN, CIFAR-100, and Tiny ImageNet 200. These are relatively similar, small-scale image datasets with few classes. The results would be much more convincing if the experiments showed generalization to at least medium-sized datasets such as ImageNet and other data modalities. Since the approach claims low overhead and requires only regular model training, there should be no reason not to run larger-scale experiments.

Similar things can be said about the models: they are all smaller-scale convolution models. It would be very interesting to see how larger-scale models behave and if the properties transfer to different architectures, e.g., transformers.

**Inconsistencies wrt. Secret k**

It is unclear to me what the purpose of the secret key k is. According to L295, it is used to make the hash values unique between different “users.” Who is the user here? I assume the attacker since they are generating the secret according to Algorithm 1. If the attacker is in control of k, how does it make anything more secure? This understanding is, however, inconsistent with L483 where the claim is that hiding the key from the user (presumably attacker?) can solve the effect of adaptive attacks.

**No Code Available**

No code was available for review, and the authors did not specify whether it would be released upon publication.

**Questions:**

Why did the authors only consider small-scale vision datasets? It seems to me like there should not be significant barriers to train on larger-scale datasets such as ImageNet.

Does this approach also work on other data modalities?

L294: What security risk does hash calculation pose? How exactly could a constant hash pose a security risk, how could it be exploited, and how do individual user keys, which are then revealed, solve these issues?

L465: Why is the attacker knowing the key k a worst-case assumption? According to Figure 2 and Algorithm 1, the attacker generates k to compute the hash and, therefore, always has knowledge of it.

L482: “Keeping the user’s key hidden from the users […] might be a solution”.

1. How can the user’s key be hidden from the user itself?
2. Who is the user here? The trainer or the verifier?
3. And how does hiding the key solve the attack?

---

> ### Author Response · Authors · 2024-11-23
> **Reply to Reviewer 5EGH (1/2)**
>
> We are deeply thankful for your detailed review and constructive feedback. Your suggestions on the threat model and framework implementation have indeed pointed out some clarity issues in our manuscript. These aspects were not fully elaborated due to space constraints. We will carefully consider your feedback and revise the corresponding sections accordingly.
>
> In the following sections, we provide detailed answers to your questions, aiming to address your concerns sufficiently, and we we kindly ask you to reconsider the evaluation of our work in light of the improvements.
>
> # $\textcolor{red}{{\rm Q.1}}$ Inconsistent Threat Model Assumptions
>
> > - ... At the same time, the threat model assumes that the same dishonest attacker **voluntarily** sticks to the exact verification protocol...
> > - In practical terms, this means it is easy for the attacker to **bypass** the proposed verification approach...
>
> **Mandatorily but not voluntarily**
>
> In our proposed TDP, we assume that the attacker's (trainer's) watermark embedding operation is mandatorily enforced and could not be modified by the trainer. We provide a detailed explanation of how this can be implemented, as well as its feasibility and rationale, in General Response $\textcolor{red}{{\rm GR.3}}$ **Implementation of TDP**
>
> **Planned Revisions**
>
> We will provide a clearer explanation of the above characteristics in the threat model and implementation section. Considering the space constraints in the main text, we may include detailed discussions on the implementation in the appendix.
>
> # $\textcolor{red}{{\rm Q.2}}$ The Dataset Has to be Public
>
> > Since the data contains a significant part of the value, the trainer may be reluctant to publish it. Licenses and intellectual property may also work against this.
>
> The nature of datasets in TDP is *Turstworthy* instead of private. We provide a detailed explanation in $\textcolor{red}{{\rm GR.2}}$.
>
> There is indeed a trade-off between dataset transparency/trustworthiness, and dataset privacy/copyright. However, the latter is a primary focus of dataset provenance, while TDP, as explored in this paper, emphasizes the former. Thus, TDP and dataset provenance are fundamentally different problems.
>
> We will incorporate your suggestion and add this discussion in the introduction or discussion section of the paper.
>
> # $\textcolor{red}{{\rm Q.3}}$ Secret Key and User
>
> **Who is the user ?**
>
> In the main text, "user" refers to the "model trainer" or, in the case of a dishonest trainer, the "attacker." We provide a detailed analysis of this in $\textcolor{red}{{\rm GR.1.1}}$.
>
> **Why the secret key is necessary ?**
>
> In $\textcolor{red}{{\rm GR.4}}$, we conduct a detailed analysis of the security of the system under both keyed-hash and non-keyed-hash scenarios. The conclusion is as follows:
>
> Keyed-hash provides **superior security** compared to common hash functions. When the details of T-Train are partially or fully exposed, only keyed-hash can enhance security by concealing the key.
>
> The effective mechanism is that even if the attacker knows the complete verification details, such as the hash function and probe types used, it remains highly difficult to brute-force and select the correct probe without knowing the key.
>
>
> **How to hide the key ?**
>
> In  $\textcolor{red}{{\rm GR.5}}$, we provide a detailed analysis of various scenarios regarding who is responsible for generating the key and propose a potential implementation scheme for situations where the key needs to be concealed. We hope this addresses your concerns.
>
> **Planned Revisions**
>
> We sincerely thank you for your valuable comments on this part. We will standardize the terminology throughout the manuscript by replacing "user" with "trainer". Considering the page constraints in the main text, we will include additional security analysis and implementation details in the appendix to support the keyed-hash mechanism in the main text.

---

> > ### Author Response · Authors · 2024-11-23
> > **Reply to Reviewer 5EGH (2/2)**
> >
> > # $\textcolor{red}{{\rm Q.4}}$ Limited Evaluation
> >
> > **Planned Revisions**
> >
> > We will incorporate your feedback to enrich and improve the experimental evaluation. Due to time constraints, we may not be able to complete and organize all the results during the rebuttal period. We sincerely apologize for this in advance, but we would try our best include these experiments in the final revised version. Meanwhile, we provide detailed responses below to address the questions.
> >
> > **larger scale dataset and model？**
> >
> > The datasets and models used in this experiment were selected based on related works (primarily those focusing on dataset provenance). Additionally, the datasets evaluated exhibit increasing complexity (SVHN → CIFAR-10 → CIFAR-100 → TinyImageNet-200). TinyImageNet, as a representative subset of the large-scale ImageNet dataset, provides some degree of support for assessing its practical effectiveness.
> >
> > Meanwhile, all experimental results in this study were computed with five repetitions, reporting both the mean and standard deviation, which already required considerable computational effort.  When expanding to larger datasets and models, we do face certain computational limitations. For instance, using the full ImageNet dataset would involve a data scale 100 times larger than the current datasets, requiring substantially more time for evaluation.
> >
> > Nonetheless, we greatly value your suggestion and will consider incorporating further improvements in the current version and future work.
> >
> > **Other data modalities?**
> >
> > We believe that, in principle, the approach can be extended to other data modalities, as overfitting and underfitting are common phenomena across deep learning systems in various modalities, enabling PP and AP to generalize effectively. However, the effectiveness of noise-based UP and TP may be less clear. We do plan to explore this as the next step of our work, and we sincerely appreciate your suggestion.
> >
> > In the current version, we intend to focus on introducing the TDP task and designing its foundational framework. Extending to other modalities would likely require addressing additional modality-specific adaptations and challenges, which we believe warrants a separate paper for a thorough and comprehensive discussion.
> >
> > # $\textcolor{red}{{\rm Q.5}}$ Code Availability
> >
> > **We commit to releasing the source code as soon as possible upon publication.**
> >
> > Currently, due to time constraints, the code still requires additional effort for organization, documentation, and preparation. As a result, it was not submitted immediately. We plan to release the code publicly after completing these refinements, along with detailed instructions to enable execution and reproduction of the experimental results.

---

> > > ### Comment · Reviewer_5EGH · 2024-11-25
> > >
> > > I thank the authors for engaging in the discussion and providing additional clarifications. However, after carefully considering all of the general and individual replies, my three main concerns remain:
> > >
> > > 1. The threat model assumes that a dishonest attacker can be forced to execute the watermarking procedure exactly as required by the protocol (Q1 / GR3). This is a very strong assumption that is hard to enforce, as the trainer is inherently in control of the code executed on their devices.
> > >
> > > 2. The proposed implementation of this assumption relies on obscuring the implementation and secret key from the attacker by providing an obfuscated binary as a wrapper to deep learning frameworks (GR3.3). However, both the implementation and the key are provided to the attacker, which enables reverse engineering and extraction. There is a long-standing principle in computer security that "security through obscurity" is a bad practice, and security should rely on well-designed mechanisms, such as strong cryptographic keys, which can remain secret. The proposed method directly violates this.
> > >
> > > 3. The authors do not provide a secure implementation of such a wrapper. They assume it to magically exist and magically be enforceable. Therefore, it cannot be analyzed and evaluated.
> > >
> > > These concerns are also shared by other reviewers. They fundamentally stop the method from doing what it is intended to. **I therefore continue to strongly argue against acceptance.**
> > >
> > > Some additional points:
> > >
> > > 4.  GR4: Only the setting DL III-B is realistic. The attacker is provided the full implementation and secret key $k$. Therefore, assuming either of those as unknown is unrealistic and dangerous.
> > >
> > > 5. Even after remarking this in my initial review, the authors are still inconsistent with how and where $k$ is generated:
> > > - According to GR3.3: "We assume the key is generated by the trainer, and the trainer has control over it."
> > > - According to GR 5: "[...] we recommend that the verifier generate and control the key [...]"

---

> > > > ### Author Response · Authors · 2024-11-27
> > > > **Thanks for Replying**
> > > >
> > > > Thank you for your reply! We will carefully consider your feedback and further improve our work.

---

### Author Response · Authors · 2024-11-23
**General Response (1/4)**

We sincerely appreciate the time and effort invested by the reviewers in providing comprehensive and insightful feedback on our submission.   The constructive comments have provided us with significant inspiration and assistance.  We will carefully consider your suggestions and continue to improve our current manuscript.  Once again, we extend our heartfelt gratitude.

In this section, we will focus on addressing the **common and key concerns** raised by the reviewers, while each specific question will be addressed in detail in our individual responses to the reviewers.

As the Trustworthy Dataset Proof (TDP) proposed in this work is a completely new problem,  many related aspects require detailed explanations to ensure clarity.   We sincerely hope that the AC and reviewers will kindly take the time to review the detailed explanations, and we are deeply grateful for your patience and understanding.


# $\textcolor{red}{{\rm GR.1}}$ Possible Real-World Applications of TDP
We would like to begin by introducing a potential application scenario that we envision for TDP:

$\textcolor{red}{{\rm GR.1.1}}$ **Verifier vs Trainer: The provider and user of the certification service**

The main interaction in the TDP framework is as follows: the trainer trains a model and submits it to the verifier for validation. In this framework, the verifier acts as a trusted entity providing a "verification service," while the trainer is the user of this service.

Upon successful verification, the verifier grants the trainer a proof of verification, which the trainer can use to enhance the credibility of their trained model. The proof effectively serves as an endorsement from the verifier.

$\textcolor{red}{{\rm GR.1.2}}$ **A possible example demostrated by Hugging Face**

Let’s use an example for better understanding: [Hugging Face](https://huggingface.co/) hosts a large number of publicly trusted datasets that anyone can download or inspect for credibility. Suppose a trainer uses the [CIFAR-10 dataset from Hugging Face](https://huggingface.co/datasets/uoft-cs/cifar10) (used here as an example, applicable to any other dataset) for unaltered training and submits the model to the verifier for validation. In this case, we assume the verification service is also provided by Hugging Face, and the verification process utilizes the trainer-declared CIFAR-10 dataset. Once the model passes verification, it can be uploaded to Hugging Face with a "Trusted-Trained on CIFAR-10" badge displayed in the [model card interface](https://huggingface.co/edadaltocg/resnet18_cifar10). Clicking on this badge could even link directly to the corresponding CIFAR-10 dataset page.

At this point, individuals looking to download the model can obtain the following information via the badge:
1. The model was trained in a trustworthy manner, endorsed by an official entity (e.g., Hugging Face in this case).
2. The dataset used for training is publicly accessible and verifiable.

This collectively enhances the model's overall credibility and transparency.

# $\textcolor{red}{{\rm GR.2}}$  The Nature of Datasets in TDP:  Turstworthy Instead of Private.

In *TDP*, we focus more on the trustworthiness of datasets, specifically addressing malicious tampering by trainers when using trusted data. Therefore, datasets in TDP should be **public** or at least **semi-public**. In the public scenario, datasets are openly available, for instance, hosted on platforms like Hugging Face or GitHub, allowing all users to independently inspect and verify their credibility. In the semi-public scenario, at a minimum, the model’s trainer and verifier must have access to the dataset. The verifier is responsible for supervising and validating the dataset's credibility and leveraging their authority to endorse it for public assurance.

In contrast, *dataset provenance* focuses more on data privacy, addressing unauthorized use of private datasets, initiated by their owners for verification. As such, these datasets are typically not made public. Notably, most current dataset provenance solutions still require dataset owners to share their private datasets with a trusted verifier for validation, thereby falling under the scenario of **semi-public**.

The nature of the dataset and the corresponding objectives of the verifier represent key distinctions between TDP and the widely-studied dataset provenance task, as the two are fundamentally different tasks. However, due to their significant similarities, confusion can easily arise. We strongly encourage the reviewers and readers to refer to  $\textcolor{green}{{\rm Figure~1}}$ in the manuscript, along with the previously introduced application scenario examples in $\textcolor{red}{{\rm GR.1.2}}$  , for a clearer understanding.

---

> ### Author Response · Authors · 2024-11-23
> **General Response (2/4)**
>
> # $\textcolor{red}{{\rm GR.3}}$  Possible Practical Implementation of TDP
>
> We first review and summarize the core verification mechanism of the verifier in TDP, followed by an analysis of how to implement the proposed Trusted Training (`T-train`) on the trainer side.
>
> $\textcolor{red}{{\rm GR.3.1}}$ **Core verification mechanism**
>
> The core principle behind the verifier’s certification process is *the detection of specific data probes within the model provided by the trainer.*
>
> $\textcolor{red}{{\rm GR.3.2}}$ **Implementation of** `T-Train`
>
> Our requirement for additional operation $\mathbb{O}$ in `T-Train` is that it must be an automated process that **cannot be manipulated by the trainer**.
>
> Here, we provide additional details that may not have been fully explained in the main text due to space constraints:
>
> ---
>
> $\textcolor{blue}{{\rm 1.Key~Initialization:}}$ We assume the key is generated by the trainer, and the trainer has control over it. For instance, the key $k$ can be generated using a specific key generation function, ${\rm KeyGen()}$.
>
> $\textcolor{blue}{{\rm 2.TrustedTraining~Wrapper:}}$ Using a commonly used deep learning framework like PyTorch as an example, the dataset/dataloader module can be wrapped using a verifier-provided wrapper module. This wrapper would automatically execute the `T-Train` process described in the paper, including probe selection and probe embedding. This is feasible because probe embedding typically involves only minor modifications to certain sample attributes, such as increasing the sampling probability of specific samples. Therefore, it is decoupled from the subsequent training optimization process and the model's framework.
>
> $\textcolor{blue}{{\rm 3. Verification:}}$ The trainer employs the aforementioned wrapper to complete model training and submits the trained model along with the key to the verifier for validation.
>
> ---
> The key prerequisite for the TDP framework to function effectively is that the wrapper must not be "compromised" by the trainer. Specifically, the trainer must not be able to manipulate the steps within the wrapper, particularly the process from handling the training dataset $D$ to probe selection.
>
> Otherwise, it is evident that a counterexample can arise: the attacker could arbitrarily tamper $D \rightarrow D^*$,  but during the probe embedding phase, directly embed probes of $D$ and claim "trained on $D$". In this way, an attacker could easily undermine the entire verification framework (as also pointed out by Reviewer $\textcolor{purple}{{\rm 5EGH}}$, $\textcolor{teal}{{\rm stcP}}$. $\textcolor{navy}{{\rm ty3A}}$). This is precisely why, in the manuscript, we emphasize in the attacker capability assumption section ($\textcolor{green}{{\rm L192}}$) that the attacker should not be able to modify the additional operation $\mathbb{O}$ (the wrapper in this case).
>
>
> $\textcolor{red}{{\rm GR.3.3}}$ **Main Concerns Regarding** `T-Train`
>
> The primary concerns that reviewers and readers might have regarding the aforementioned wrapper can be summarized as follows:
>
> $\textcolor{blue}{{\rm Q1.}}$ **How to ensure that the trainer is compelled to use the wrapper in** `T-Train`  **?**
>
> $\textcolor{green}{{\rm A1.}}$ Due to the presence of the verification mechanism, only models with correctly embedded probes can pass the verification. Therefore, if the trainer wishes to utilize the verification service, he must use the trusted training wrapper.
>
> $\textcolor{blue}{{\rm Q2.}}$ **How to ensure that the trainer cannot compromise the wrapper in** `T-Train`  **?**
>
> $\textcolor{green}{{\rm A2.}}$ This is perhaps the most challenging step in implementing the framework, especially considering the scenario where the wrapper's code is entirely open source. However, we believe that leveraging **closed-source techniques** can address this issue. For example, critical logic can be placed in a closed-source dynamic library (e.g., `.dll` or `.so` files) and invoked via Python’s `ctypes` module. Alternatively, the key logic could be implemented in `Cython`, compiled into a binary file, and called accordingly. Additionally, tools such as `PyArmor` can be used to encrypt or obfuscate the implementation details, ensuring the protection of the underlying logic.
>
> Based on the above analysis, we believe it is a reasonable and achievable assumption that the trainer must execute the wrapper as-is and cannot modify it.

---

> ### Author Response · Authors · 2024-11-23
> **General Response (3/4)**
>
> # $\textcolor{red}{{\rm GR.4}}$ Detailed Security Analysis of TDP
>
> $\textcolor{red}{{\rm GR.4.1}}$ **Analysis of different dangerous levels**
>
> Based on the reviewers' questions, we categorize the security of TDP into three $\textcolor{blue}{{\rm Dangerous~Levels (DL):}}$  depending on the malicious trainer's knowledge of the verification mechanism. For each Level, we discuss two cases: **(A)** the attacker does not know their key, and **(B)** the attacker knows their key. Additionally, we present case **(C)**, where a keyed-hash is not used, for comparison. The comprehensive comparison is illustrated in [THIS CHART](https://smms.app/image/cljwNiFZ9ghYaUL) ($\leftarrow$ recommend clicking the provided link to view it).
>
> ---
>
> $\textcolor{blue}{{\rm DL~I:}}$  **No-info**. The attacker is completely unaware of the details of the verification mechanism. Evidently, **I-A**, **I-B**, and **I-C** are secure.
>
> $\textcolor{blue}{{\rm DL~II:}}$  **Partial-info**. The trainer has partial knowledge of the verification mechanism, such as knowing about the hash-based data probe embedding strategy but not the specific hash function used.
> At this point, the attacker may adopt a *brute-force* strategy, such as repeatedly trying different hash functions in an attempt to guess the actual probe. Thus, both **II-B** and **II-C** carry certain security risks. However, if the key remains inaccessible to the attacker (**II-A**), security can still be ensured.
>
> $\textcolor{blue}{{\rm DL~III:}}$  **Full-info**. This is the worst-case scenario, where the attacker is fully aware of all the details of the verification mechanism. In **III-B** and **III-C**, as noted by the reviewers, the attacker could indeed adopt a more thorough attack strategy: bypassing the aforementioned wrapper entirely and directly embedding probes of $D$ during training, thereby undermining the verification mechanism. However, this assumption grants the attacker *excessive power* and essentially equates to allowing them to manipulate the Trusted Training process (refer to the discussion in $\textcolor{red}{{\rm GR.3.2}}$). Nevertheless, if the trainer does not know the key (**III-A**), it remains highly unlikely for them to select the correct probe, thereby ensuring security.
>
> ---
>
> $\textcolor{red}{{\rm GR.4.2}}$  **Significance of the Proposed Adaptive Attack**
>
> In our experimental evaluation, we designed a Forge Probe Attack **(FPA)** ($\textcolor{green}{{\rm L465}}$). After considering the reviewers' feedback, we recognize that certain parts of the discussion were indeed unclear. We will address and clarify these issues in the revised version.
>
> FPA should not correspond to the **III-B** scenario, as this grants the attacker overly strong assumptions and, to some extent, contradicts the assumptions of this work. Instead, FPA corresponds to  **II-B** and similar scenarios, where the attacker may rely on brute-force strategies to infer the actual probes. In this case, fully training the model while attempting different probe would be highly time-consuming. In contrast, **FPA provides a highly efficient attack approach**, as it only fine-tunes the model on probe samples (e.g., approximately 1% of the original dataset size).  This makes it more feasible for the attacker to brute-force the correct result. Therefore, $\textcolor{green}{{\rm Table~3}}$ in the manuscript essentially demonstrates the robustness of the system when the attacker guesses correctly.
>
> In summary, we draw the following two conclusions:
> 1. **Keyed-hash outperforms common hash functions.** While the security of both is identical when the trainer knows the key, in higher-risk scenarios (II and III), the keyed-hash provides an additional option to enhance security by concealing the key.
> 2. **Hiding the key from attacker improves the security.** Our experiments on the FPA attack demonstrate that attackers (trainers) can employ a quick testing strategy to guess probes, with PP and AP probes facing higher risks. Concealing the key significantly expands the search space, making it much harder for the attacker to guess correctly.
>
> Additionally, more complex hash function strategies can be designed (e.g., combining the results of two different hash algorithms following specific rules) or new probe types can be developed to effectively increase the search space and reduce the likelihood of attackers deducing the details of the verification mechanism.

---

> ### Author Response · Authors · 2024-11-23
> **General Response (4/4)**
>
> # $\textcolor{red}{{\rm GR.5}}$  Who Generates the Key $k$?
>
> In the manuscript, $\textcolor{green}{{\rm Figure~3}}$ illustrates only the demonstrates the principles of the framework, highlighting that a key $k$ needs to be generated during the `T-Train` process.
>
> From an implementation perspective, based on the analysis above, we can see that when the **security risk is low**, the trainer can generate the key themselves. For instance, the verifyer provide a ${\rm KeyGen()}$ function. Each trainer is assigned a unique ID $d$, and it could excute ${\rm k \leftarrow KeyGen(d)}$, ensuring that $k$ meets specific formatting requirements.
>
> When **security risks are present** (as analyzed in $\textcolor{red}{{\rm GR.4}}$ under scenarios DL-II and DL-III), we recommend that the verifier generate and control the key, ensuring it is not accessible to the trainer.  This is precisely the meaning of *"through a server API"* mentioned in the paper ($\textcolor{green}{{\rm L485}}$). Building on the implementation methods discussed in $\textcolor{red}{{\rm GR.3.2}}$, we propose a potential implementation scheme:
>
> ---
>
> $\textcolor{blue}{{\rm 1.Key~Initialization:}}$  The trainer first registers on the server provided by the verifier and only obtains a user ID $d$. The verifier then generate the key ${\rm k \leftarrow KeyGen(d)}$ and retains the pair $<d, k>$  exclusively on the server (verifier) side.
>
> $\textcolor{blue}{{\rm 2.TrustedTraining~Wrapper:}}$  The trainer inputs its ID $d$ (not the  key $k$) into the wrapper. The wrapper sends $d$ to the server, which returns the key $k$. The wrapper then proceeds keyed-hash and the subsequent processes.
>
> $\textcolor{blue}{{\rm 3. Verification:}}$
> The trainer submits its ID $d$ to the verifier, which then retrieves the corresponding key $k$ from the stored pair $<d, k>$ on the server and proceeds with the subsequent verification process.
>
> ---
>
> Note that the communication process in the wrapper must also adhere to the assumptions of this work: it must not be "compromised" by the trainer. Specifically, the trainer should not be able to intercept or decrypt the key.

---

### Note · Authors · 2024-11-27

I have read and agree with the venue's withdrawal policy on behalf of myself and my co-authors.